# On the relationship of energetic particle precipitation and mesopause temperature

Florine Enengl[1,2,3,4], Noora Partamies[2,5], Nickolay Ivchenko[1], and Lisa Baddeley[2,5]

[1]KTH Royal Insitute of Technology, Stockholm, Sweden
[2]The University Centre in Svalbard, Norway
[3]ESA European Space Research and Technology Centre, The Netherlands
[4]University of Oslo, Norway
[5]Birkeland Centre for Space Science, Norway

**Correspondence:** Florine Enengl (florine@kth.se)

**Abstract.**

Energetic particle precipitation (EPP) has the potential to change the neutral atmospheric temperature at the mesopause region. However, recent results are inconsistent leaving the mechanism and the actual effect still unresolved. In this study we have searched for electron precipitation events and investigated a possible correlation between D region electron density en-
hancements and simultaneous neutral temperature changes. The rotational temperature of the excited hydroxyl (OH) molecules is retrieved from the infrared spectrum of the OH airglow.

The electron density is monitored by the European Incoherent Scatter Scientific Association (EISCAT) Svalbard Radar. We use all available experiments from the International Polar Year (IPY) in 2007–2008 until February 2019. Particle precipitation events are characterized by rapid increases in electron density by a factor of 4 at an altitude range of 80–95 km, which
overlaps with the nominal altitude of the infrared OH airglow layer. The OH airglow measurements and the electron density measurements are co-located. Six of the ten analysed electron precipitation events are associated with a temperature decrease of 10–20 K. Four events were related to temperature change less than 10 K.

We interpret the results in terms of the change in the chemical composition in the mesosphere. Due to EPP ionisation the population of excited OH at the top of the airglow layer may decrease. As a consequence, the airglow peak height changes
and the temperatures are probed at lower altitudes. The observed change in temperature thus depends on the behaviour of the vertical temperature profile within the airglow layer. This is in agreement with conclusions of earlier studies, but is, for the first time, constructed from electron precipitation measurements as opposed to proxies. The EPP related temperature change recovers very fast, typically within less than 60 minutes. We therefore further conclude that this type of EPP event reaching the mesopause region would only have a significant impact on the longer-term heat balance in the mesosphere if the lifetime
of the precipitation was much longer than that of an EPP event (30–60min) found in this study.

# 1 Introduction

Space weather phenomena can affect the dynamics and the heat balance of the atmosphere by depositing energy in the form of energetic particle precipitation (EPP). In particular, investigating the mesopause region (at 80–100 km), the boundary between the mesosphere and the thermosphere (Andrews, 2010), is important. Within this boundary, the behaviour of neutral gas and ionized particles differ, which is why complex interactions between dynamics, photochemistry, heating and transport mechanisms take place and the atmospheric energy budget can be altered.

Several studies have investigated the effects of EPP on neutral temperatures in the mesopause region. Nesse Tyssøy et al. (2010) compared particle precipitation observed by the NOAA (the National Oceanic and Atmospheric Administration) POES (Polar Orbiting Environmental Satellites) satellite's MEPED (Medium Energy Proton and Electron Detectors) instrument with neutral temperatures derived from the TIMED (Thermosphere, Ionosphere, Mesosphere, Energetics and Dynamics) satellite's SABER (Sounding of the Atmosphere using Broadband Emission Radiometry) instrument. They used a dataset of 80 days within May–June and October–November in 2003 over the northern hemisphere. Temperature profiles were averaged over all local time hours for four flux levels of precipitating energetic protons. A temperature increase of about 40 K was found at 115–120 km associated with strong fluxes of 80–250 keV protons in October–November (northern hemispheric early winter). The strongest temperature increase of 15–20 K in May–June (northern hemispheric early summer) at the altitudes of 110–115 km was reported due to high fluxes of 30–80 keV protons. As particle precipitation events change the Pedersen conductivity, Joule heating may contribute to the increase in temperature at these altitudes. Further down in the atmosphere, at 85–90 km, a cooling of 3–4 K during periods of high Kp values was observed due to precipitation of 250–800 keV protons (an intense solar proton event in the period of 27 October – 6 November 2003).

The rotational hydroxyl (OH) airglow temperature was observed during six nights of the austral winter in 2008 by Suzuki et al. (2010b). These nights were selected to include high auroral activity and clear weather. In their study, the temperature was derived from spectra recorded at Syowa Station in Antarctica by a high-sensitivity spectrometer. During only one of these nights (27–28 March 2008), an increase in the temperature of 10 K over a time period of 15 minutes was observed. Furthermore, a decrease in the relative intensity of the OH(8–4) Q branch of ∼23% was found by comparing the pre-EPP level to that half an hour after the lowest local minimum of the horizontal H component. No such coherent behaviour was seen during the other nights. Suzuki et al. (2010b) suggested a relationship between EPP and the OH temperature based on measured disturbances in the horizontal magnetic field and variations in the cosmic radio noise absorption (CNA) over the course of several hours of geomagnetic activity. The average energy of the precipitating electrons during this night reached 10–20 keV. They discussed different mechanisms for causing the change in the temperature. Joule heating was concluded to not contribute much, as the estimated heating rate required to explain the observed temperature increase at the mesopause height was three orders of magnitude higher than the heating rate the particle precipitation observations suggested. Direct particle heating, on the other hand, is produced by precipitating particles colliding with the atmospheric neutrals. This heating process is dependent on the incident particle energy and their deposition altitude. Similarly to the Joule heating, the authors estimated the energy deposition rate of EPP at the mesopause height, and concluded that the high-energy particle flux required

to explain the temperature change was unrealistically high. Atmospheric gravity waves were also excluded, as the intensity of the OH airglow and the rotational temperature did not show a positive correlation, which is characteristic for dynamics driven conditions, as described and modelled by Cho and Shepherd (2006). Suzuki et al. (2010b) further discussed the possibility of a change in the height distribution of the OH airglow emission during auroral events. The initial profile of OH volume emission rates retrieved by the SABER instrument onboard the TIMED satellite was compared to an example of the disturbed layer. The comparison showed a decrease in the upper part of the disturbed layer as the thickness of the OH layer had decreased by 20%. If a change in the height distribution of the OH airglow emission occurred during an auroral event, an increase in temperature is not necessarily observed, as the outcome would depend on the local temperature in the mesosphere.

The connection between the geomagnetic activity and the long-term temperature at the mesopause region during solar cycles 23 and 24 was studied by Gavrilyeva and Ammosov (2018). The OH rotational temperature was measured by the ground-based infrared spectrograph at Maimaga station (63°N, 129.5°E), and ascribed to an altitude of 87 km, which is commonly assumed to be the peak height of the OH layer. The seasonally averaged temperatures from 1999 to 2015 were included in the analysis. The results showed that the winter mesopause temperature from October to February is about 10 K higher during the years with high geomagnetic activity (Ap > 8) than during low geomagnetic activity years (Ap ≤ 8). This suggests a correlation between the mesopause temperatures and the solar activity. A more direct solar activity dependence of OH airglow temperatures was reported by Holmen et al. (2014). They concluded on a temperature change of about 4 K per 100 solar flux units (SFU) of the F10.7 radio flux. The question on the short-term relationship between EPP and the mesopause temperature was still left open.

The purpose of this study is to characterize the effect of the EPP on the mesopause temperature in more detail, using co-located measurements of precipitating electrons and the mesospheric temperature. The instrumentation is further described in Section 2. Section 3 outlines the data used in this study, as well as the analysis of the EISCAT Svalbard Radar data. Finally, the results are shown and discussed in Sections 4 and 5. The conclusions of our findings are summarized in Section 6.

## 2   Instrumentation

Following Cresswell-Moorcock et al. (2013) we use the European Incoherent Scatter Scientific Association (EISCAT) Svalbard radar data to identify electron precipitation events as electron density enhancements. Simultaneous and co-located neutral temperature measurements are determined from OH airglow spectra collected by a spectrometer. The derived rotational OH temperatures are taken as the neutral temperature of the mesopause region, assuming that the excited OH molecules are in thermal equilibrium with the ambient atmosphere.

### 2.1   EISCAT Svalbard Radar

For this study the EISCAT Svalbard Radar (ESR, Wannberg et al. (1997)) in Longyearbyen, Norway (situated at a geographic latitude of 78.15°N and a geographic longitude of 16.02°E and at corrected geomagnetic coordinates of 75.43°N and 110.68°E) is used. The radar operates at the 500 MHz band and has a 32 m steerable parabolic dish antenna and a 42 m fixed parabolic antenna aligned with the local geomagnetic field. For the purpose of this study, we searched for experiments with good height

resolution (less than 5 km) at mesopause altitudes. The `manda` experiment (Tjulin, 2017) resolves altitudes of 80 to 100 km with 1–2 km height resolution and was therefore chosen for the radar campaign in January and February 2019 for this study. In addition to our experiment and all previously run `manda`, all previous `ipy` experiments were also investigated, as `ipy` covers the mesopause region with a vertical resolution of 4–5 km. The `manda` experiments utilize the 32 m dish and the `ipy` data are collected on the 42 m dish.

## 2.2   Ebert-Fastie airglow spectrometer

The Ebert-Fastie spectrometer at the Kjell Henriksen Observatory (KHO) in Longyearbyen is used to retrieve the winter temperature of the mesopause (Sigernes et al., 2003). The observatory is located only a kilometer away from the radar site, so it is practically co-located. The spectrometer scans the near-infrared wavelength region from 824 to 871 nm which includes the rotational-vibrational OH(6–2) band of the OH airglow. The spectrometer points to the zenith with a field-of-view of 5 degrees. It measures whenever the Sun is more than 12 degrees below the horizon, which at KHO latitude (78.2°N) gives an optical observation season from the beginning of November until the end of February. The spectral resolution of the spectrometer measurement of the OH(6–2) band is 0.4 nm. One scan takes 25 seconds but to obtain a good signal-to-noise ratio many scans are averaged during post-processing of the data. Most earlier studies use 1-hour averages (144 scans). In this study, half-hour averaging (72 scans) is used for better temporal resolution. Improving the temporal resolution reduces SNR, which is reflected by larger temperature errors. It is justified by the aim to study short-term temperature changes. For the events analysed in this study, the selection of "good" data still uses the same criteria that has earlier been applied to hourly averaged data. We further move the 30-min averaging window by 10-min time steps to obtain a smoothed temporal evolution of temperature. The rotational OH temperatures are obtained by fitting a synthetic spectrum to the measured band of emission lines. Using the intensities of four emission line pairs ($P_1$ & $P_2$) within the P branch of OH(6–2) the slope of the best fit determines the temperature. An auroral emission line from atomic oxygen at 844.6 nm lies in the middle of the OH(6–2) spectrum. When that emission intensity overtakes the OH emission intensity (fit covariance greater than 0.5) are excluded in the temperature analysis due an inaccurate fit. Other things causing poor fits and missing temperature values are cloudiness, high background illumination (e.g. scattered moonlight) or technical issues with the instrument. The threshold values for the fit variances have been determined empirically by viewing and fitting large datasets over decades. For consistency, at the event selection state we have employed the same threshold values as the earlier work by Sigernes et al. (2003); Holmen et al. (2013). Once the EPP events were selected, we relaxed the threshold values to obtain a more continuous data coverage. The covariance threshold was lowered to 0.2 as long as the fit variance criteria were still met. Similarly, the fit variance threshold value for $P_2$ lines was increased from 0.3 to 1.0 as long as the covariance ($< 0.5$) and $P_1$ line ($< 0.05$) criteria held. The uncertainties introduced by this procedure are reflected by the error bars in the displayed data. The accuracy of the fitting method in estimating rotational temperatures is $\pm 2$ K. The temperature error corresponds to the Boltzmann plot least square fit error, and are typically somewhat larger than the uncertainty of the method itself (see values in Table 1). In addition to the fitted temperature values, we use relative band brightness (non-calibrated arbitrary units) of the P branch of the OH(6–2) transition. These intensity values are routinely calculated and saved together with the rotational temperatures.

## 3 Data description and event selection

A total of 10220 hours of ESR data were initially inspected. The `ipy` and `manda` experiments contribute 10144 hours and 76 hours of data respectively. These experiments provide a sufficient height resolution (<5 km) to detect enhanced electron densities in the mesopause region. Here the mesopause region is defined from an altitude range of 80 to 100 km to include the hydroxyl (OH) peak height at 76–90 km (Mulligan et al., 2009). The chosen experiments are run either in the geomagnetic field-aligned direction (42m: `ipy`, 32m: `manda`) or with vertical pointing (32m: `manda`). The data set starts from the International Polar Year (IPY) in 2007–2008 when ESR was run continuously from 1 March 2007 to 29 February 2008 (Blelly et al., 2010). This IPY year includes 8784 hours of `ipy` experiment. Additionally, all `manda` and `ipy` experiments from December, January and February since the IPY year until February 2019 are included in the event search. The total of 1388 hours of ESR data between 2008–2019 was used to search for EPP events. Finally, the ESIRI experiment (ESR Ionospheric D-Region Experiment for Investigation of EPP) `manda` mode was run for a total of 48 hours in January (24 hours) and February 2019 (another 24 hours) to specifically collect data for this study. The experiment was run for six evenings between 16 to 22 UT (19 to 01 MLT).

The EISCAT raw data files (Auto-Correlation Functions (ACFs)) are analysed using the GUISDAP data package (Lehtinen and Huuskonen, 1996). This provides an iterative fitting of the ACFs to produce the electron densities examined in this study. The post-integration time is 60 s. Data gaps are shown by white areas in the plots (Figures 1 and 2) and occur where GUISDAP has not managed to provide a fit. 10-minute averaged electron densities are calculated for the altitude ranges of 87–90 km and 91–94 km separately. The electron density measurements become too noisy towards the lower part of the airglow layer, so we chose to only include the heights overlapping with the top part of the airglow layer. The error of the electron density is averaged to give a mean error for further analysis.

The search for the EPP events in the radar data is based on an earlier study by Cresswell-Moorcock et al. (2013), in which onsets were found as sudden increases (a factor 5 over 5 minutes) of electron density median values. For this study the electron density is averaged to 10 min resolution with an altitude resolution of 4 km. The previous study's factor of 5 was attempted but since there were a number of events just below this threshold the factor was adjusted to 4. We thus searched for an increase of electron density by a factor of 4 within 20 minutes. If onsets were found only 10 minutes apart, the latter onset was ignored.

The events were detected automatically but inspected and classified visually based on the temporal evolution of the electron density and the OH temperature. Events with large electron density errors of more than $5 \times 10^9$ m$^{-3}$ were excluded. Events which lack more than one temperature measurement at or after the onset were also excluded. Events which show a particle precipitation signature of electron density enhancement through the whole ionospheric column (87–126 km) were sorted into the EPP category. Events which show an electron density enhancement limited to horizontal layer at the bottom of the E region (around 100 km) were categorized as sporadic E layers (Rapp et al., 2011). The remaining data not showing clear EPP or sporadic E layer behaviour were excluded from the analysis. The sporadic E layer events are not discussed further in this paper. This selection process resulted in a set of ten EPP events, which were categorized into groups of increasing, decreasing and stable (no change) temperature evolution over the EPP onset.

The electron density during the radar run on 6 January 2019 from 16:00 UT to 22:00 UT is shown in Figure 1 (top panel). The onset of the EPP took place at 19:50 UT and the event lasted until 20:20 UT. The background electron density at the lower part of the ionosphere (below 100 km) prior to the EPP onset was low (below $\sim 10^{10}$ m$^{-3}$) but abruptly increased at the EPP onset time from the pre-EPP value to a local maximum in 10 minutes. The OH rotational temperature (middle panel) experiences large variations ($\sim$20 K) during the hours leading to the precipitation onset when there is no electron density enhancements below 100 km altitude. A small temperature dip ($\sim$5 K) is seen 10 minutes after the onset, but the change is well within the measurement uncertainties and does not persist. The relative OH band brightness (bottom panel) minimised half an hour before the precipitation started and remained at that level until half an hour after the EEP onset. During this experiment the radar was pointed to zenith.

## 4   Results

Our ten EPP events are listed in Table 1. The criterion for a changing mesopause temperature is that the change is larger than the uncertainty of the temperatures averaged over half an hour. However, following the analysis by Suzuki et al. (2010b) where temperature variability less than 10 K is considered minor and that larger than 10 K is considered significant, we additionally include the *second event* in the class of "decreasing" temperatures, despite the large uncertainties. An event is classified as stable (no change), if the temperature change from the value just before the onset to that 20 min after the onset time is a minor temperature change and within the uncertainty of the averaged temperatures. As each of the data points is an average over the previous 30 minutes the point just prior to the onset is a representative for the temperature level before the precipitation starts, while the temperature value 20 minutes after the onset includes onset time information as well as any immediate changes after the onset. Apart from the *ninth* and *tenth* event, all other events were measured by a field-aligned pointing radar experiment. Thus, the auroral emission does not obscure the temperature calculations as the two parameters were often measured side by side rather than in exactly the same column. The events associated with a temperature decrease (6 events) show a change from 10 to 40 degrees while events with a temperature increase (4 events) only undergo a few degrees' change and are thus classified as stable.

| # | Event Date [UT] | $T_{-2}$ [K] | $T_{-1}$ [K] | $T_0$ [K] | $T_{+1}$ [K] | $T_{+2}$ [K] | $\triangle T$ [K] | classification |
|---|-----------------|-------------|-------------|-----------|--------------|--------------|------------------|----------------|
| 1 | 2007/12/29 23:30 | $225 \pm 6$ | | | $188 \pm 16$ | $185 \pm 12$ | -40 | decreasing |
| 2 | 2008/02/28 02:50 | $227 \pm 10$ | $215 \pm 21$ | $187 \pm 18$ | $207 \pm 6$ | $202 \pm 21$ | -13 | decreasing |
| 3 | 2013/12/06 21:50 | $224 \pm 9$ | $219 \pm 3$ | $212 \pm 4$ | $210 \pm 0$ | $196 \pm 3$ | -23 | decreasing |
| 4 | 2014/01/21 15:20 | $229 \pm 8$ | $222 \pm 2$ | $214 \pm 4$ | $211 \pm 4$ | $212 \pm 5$ | -10 | decreasing |
| 5 | 2014/01/24 02:10 | $212 \pm 5$ | $210 \pm 5$ | $198 \pm 2$ | $199 \pm 3$ | $198 \pm 2$ | -12 | decreasing |
| 6 | 2014/01/24 15:20 | $218 \pm 14$ | $214 \pm 9$ | $212 \pm 9$ | $217 \pm 7$ | $217 \pm 11$ | +3 | stable |
| 7 | 2014/01/25 02:40 | $228 \pm 6$ | $221 \pm 4$ | $224 \pm 2$ | $221 \pm 5$ | $224 \pm 3$ | +3 | stable |
| 8 | 2016/02/06 19:50 | $194 \pm 4$ | $195 \pm 7$ | $187 \pm 10$ | $184 \pm 6$ | $182 \pm 4$ | -13 | decreasing |
| 9 | 2019/01/05 21:50 | $202 \pm 3$ | $203 \pm 8$ | $198 \pm 5$ | $194 \pm 7$ | $210 \pm 7$ | +7 | stable |
| 10 | 2019/01/06 19:50 | $196 \pm 6$ | $198 \pm 3$ | $192 \pm 3$ | $188 \pm 5$ | $199 \pm 5$ | +1 | stable |

**Table 1.** Airglow temperature values (degrees in Kelvin in 30 min averages) before, at and after the event onsets for each EPP event. $T_0$ indicates the point closest to the event onset. The data points are ten minutes apart. For instance, $T_{-1}$ refers to the data point 10 minutes before $T_0$. All temperature values are accompanied with the fitting error. The last columns indicate the observed change in the temperature over the EPP onset. The points compared in the $\triangle T$ column are $T_{-1}$ to $T_{+2}$ (as $T_{-1}$ is not available for the first event $T_{-2}$ used).

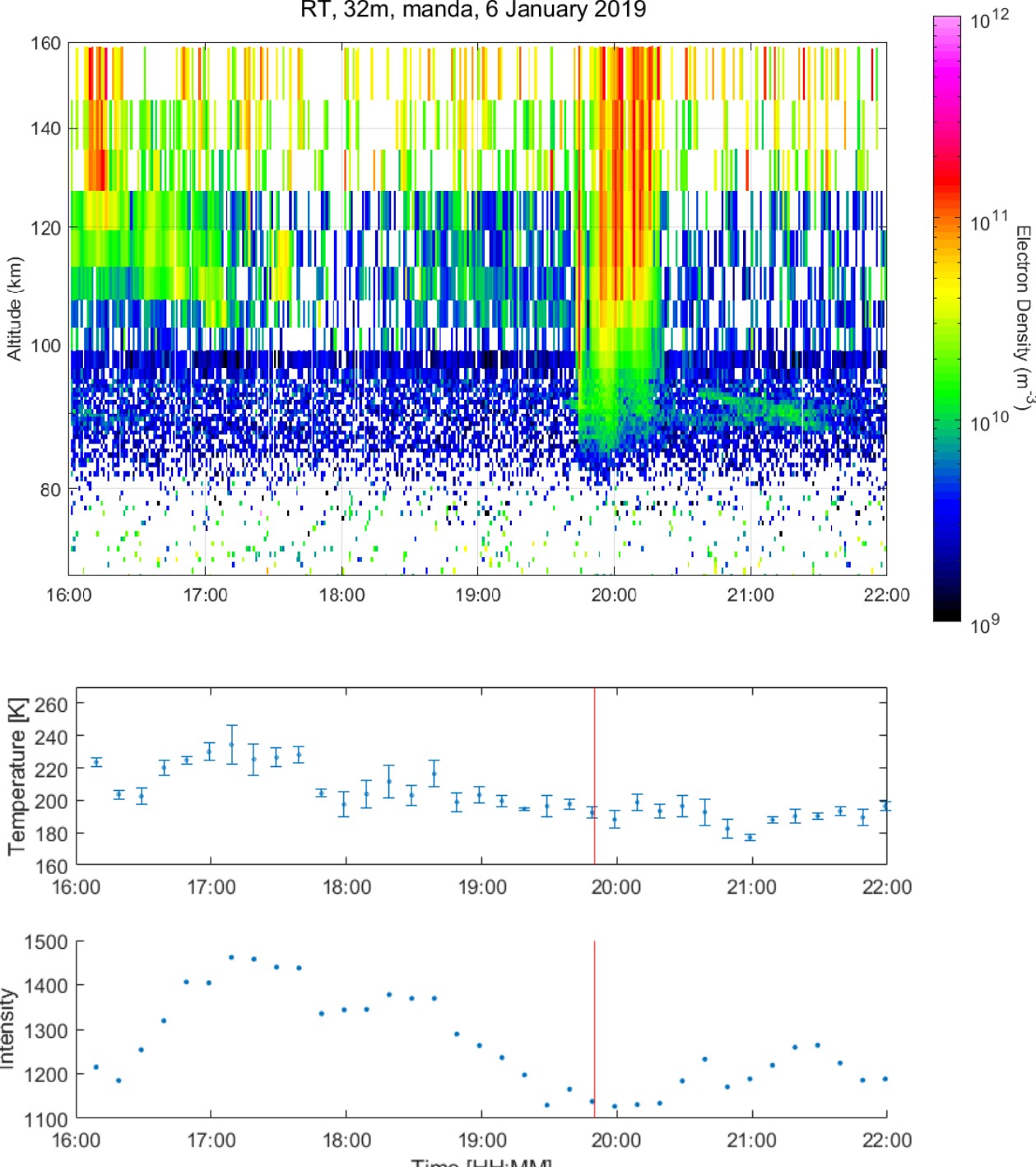

**Figure 1.** The upper panel shows the electron density as a function of altitude and time for the `manda` experiment at 16:00–22:00 UT on 6 January 2019 (*tenth event*). Particle precipitation event starts at 19:50 UT (vertical red line). The data are post-integrated to 0.6 km resolution at 80 km height to 3–4 km resolution at 100 km height and 60 s time resolution. The middle panel contains OH temperature data and the bottom panel displays the evolution of the relative band brightness, both in 30-minute averages at every 10 minute.

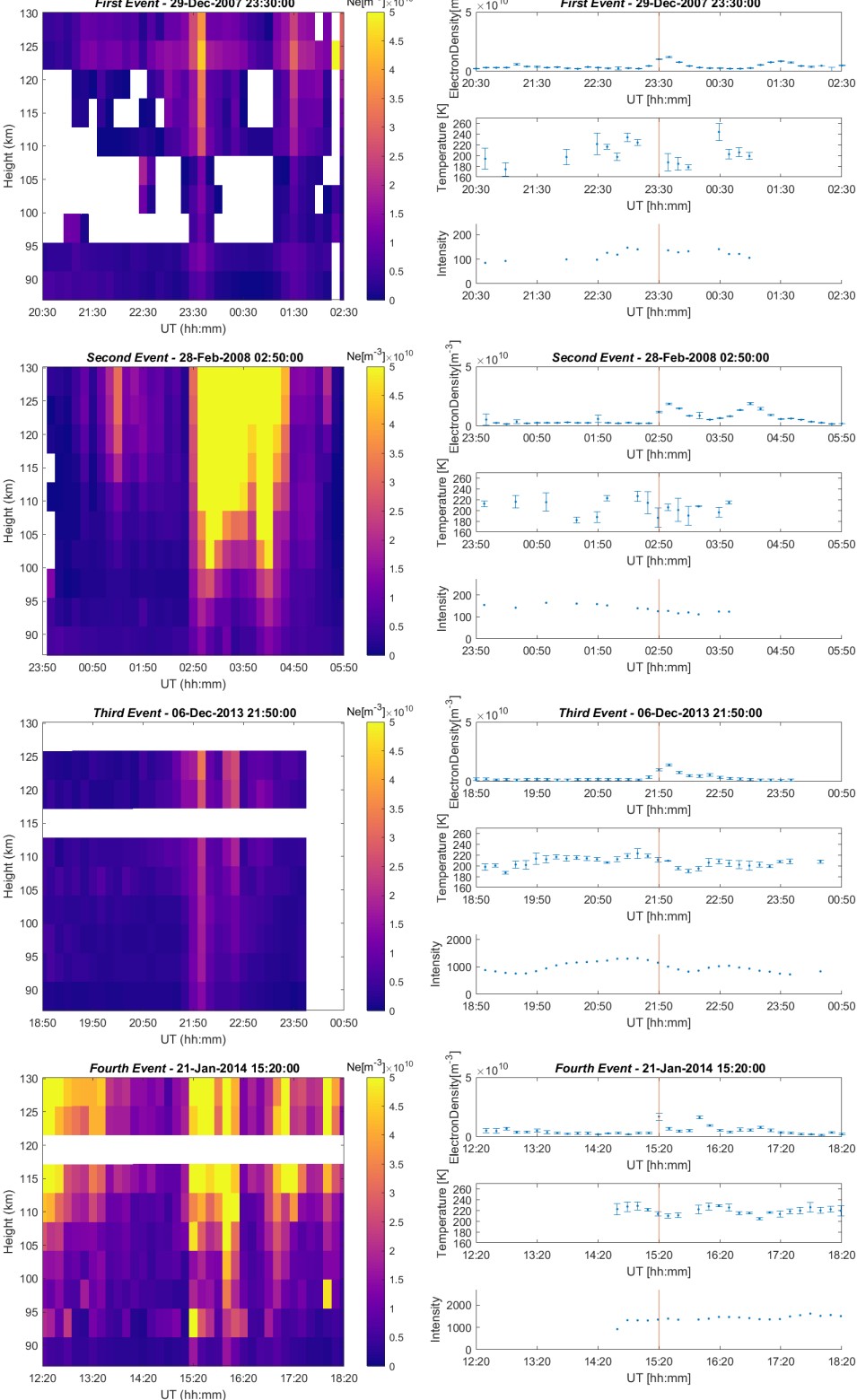

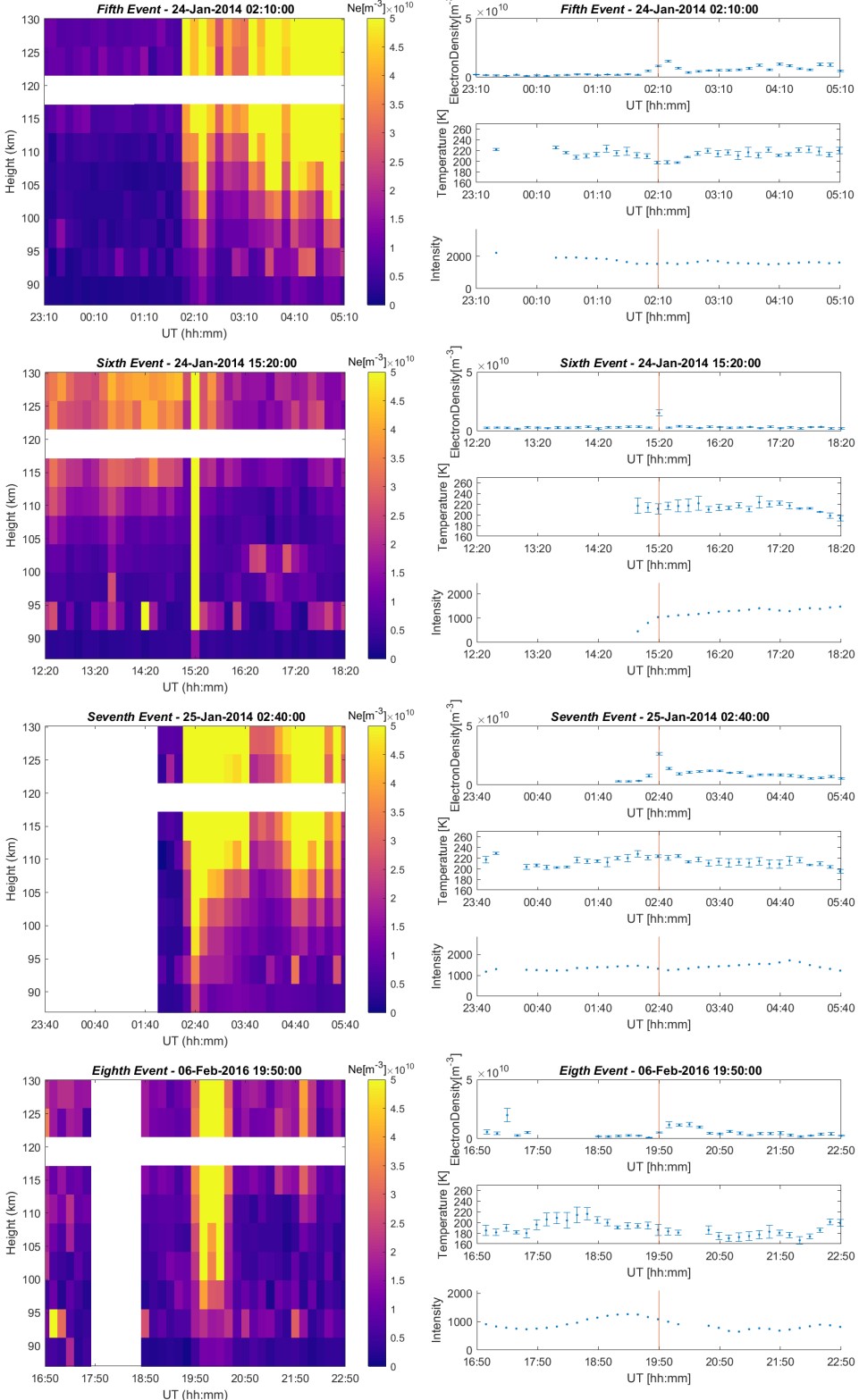

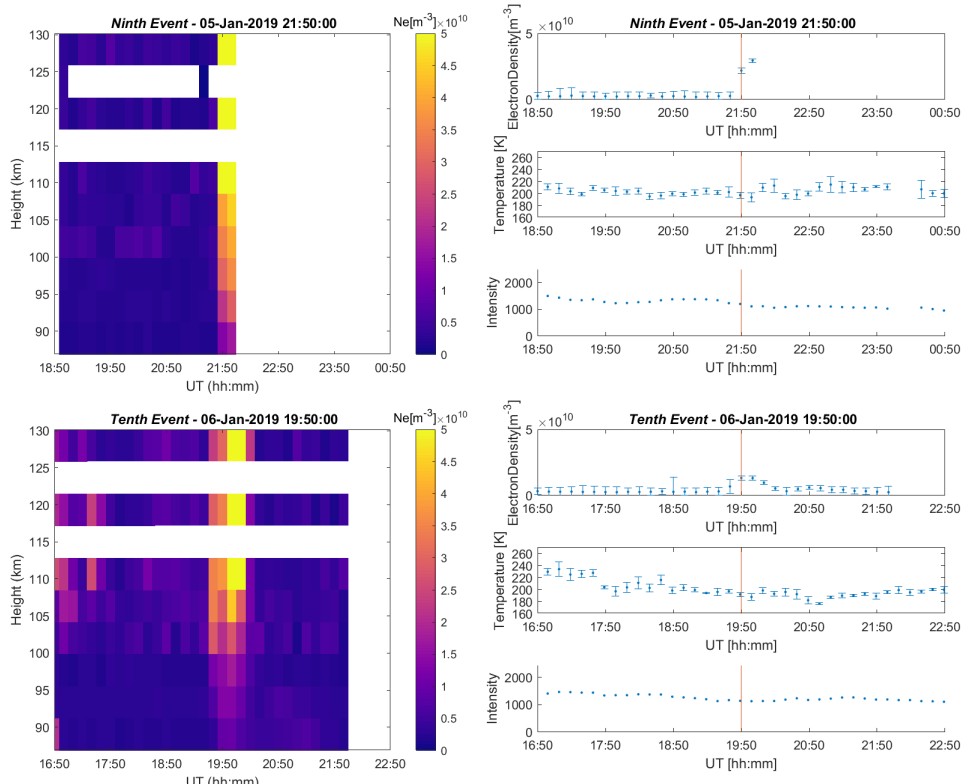

**Figure 2.** The left column shows the averaged electron density as a function of altitude and time. The right column illustrates the electron density time evolution at the height range where the electron density enhancement was detected. The temporal evolution of the airglow temperature (middle panel) and the relative OH(6–2) band intensity in arbitrary units (bottom panel) are also included. The temperature and intensity measurements are shown at the end of their half-hour integration time. All time axes are given from 3 hours before to 3 hours after the EPP onset, which takes place in the middle of the plot. The white gaps in the electron density plots indicate missing data.

Figure 2 displays each individual EPP event. The left panels present electron density measurements as a function of time and altitude, for a 6 hour interval centered at the EPP onset. The right column includes the temporal evolution of the D region electron density (top panel), the airglow temperature (middle panel) and the relative OH(6–2) band intensity (bottom panel). In the line plots the electron densities are from the lowest height range where the increase by a factor 4 was detected. Brief descriptions of each event are given in the following paragraphs.

*The first event* commences at 23:30 UT on 29 December 2007 and is detected in the altitude range of 91–94 km. The electron density at the EPP event start time is $1.0 \times 10^{10}$ $\mathrm{m}^{-3}$. The increased precipitation electron flux lasts for about 30 min. The mesopause temperature decreases by 40 K (from 225 to 185 K) over the EPP onset time, and recovers within an hour after the onset. Here the T-1 temperature point is missing. Therefore the previous point is used as a pre-EPP reference. The relative OH(6–2) band intensity is 148 before the event and diminishes to 129 over the EPP onset.

*The second event* starts at 02:50 UT on 28 February 2008. The electron density enhancement is detected in the height range of 91–94 km with an onset time value of $1.2 \times 10^{10}$ m$^{-3}$. The precipitation lasts for about 40 min and is followed by another increase lasting for about 50 min (from 04:00 until 04:50 UT). An airglow temperature decrease of 13 K (from 215 K to 202 K) is observed at the EPP onset time. The decrease is large but still within the error bars. The relative OH(6–2) band intensity decreases gradually from 2 hours before the event until the event onset time. Thus, during this event the intensity behavior is much smoother than that of the temperature. Little or no correlation is seen between the two parameters.

*The third* event starts at 21:50 UT on 6 December 2013. It is detected in the altitude range of 87–90 km with an electron density value of $9.9 \times 10^9$ m$^{-3}$. The temperature decreases by 23 K (from 219 to 196 K) over the event onset time. Both temperature and electron density values recover within an hour. The relative band intensity peaks at 1320 half an hour before the onset, decreases to 1011 at the onset and minimizes at 864 after the electron density maximum. A positive correlation between the temperature and band intensity is found in this case.

*The fourth event* starts at 15:20 UT on 21 January 2014, detected in the altitude range of 87–90 km with an electron density value of $1.7 \times 10^{10}$ m$^{-3}$. The precipitation continues for about an hour. The temperature falls by 10 K (from 222 to 212 K) over the onset time. The electron density falls within the next half hour but increases again to $1.6 \times 10^{10}$ m$^{-3}$ about 40 minutes after the onset. The temperature values increase to its background level at the time the electron density peaks a second time. The relative band intensity is stable. However the slight intensity variations coincide with the peaks and drops in the temperature evolution.

*The fifth event* starts at 02:10 UT on 24 January 2014 at an altitude range of 87–90 km. The electron density at the precipitation onset is $9.5 \times 10^9$ m$^{-3}$, and stays elevated for about two hours after the initial enhancement, which only lasts for about 30 minutes. The mesopause temperature undergoes a decrease of about 12 K, from 210 K prior to the event to 198 K after the EPP onset. The temperature recovery follows that of the initial enhancement in electron density. The OH(6–2) band intensity decreases from 1890 to 1551 already an hour prior to the EPP onset. Similar to the temperature evolution, the emission intensity recovers within half an hour after the onset. There is a mild positive correlation between the temperature and band intensity values.

*The sixth event* commences at 15:20 UT on 24 January 2014 in the altitude range of 87–90 km. The electron density enhancement is seen in a single profile only (10 min lifetime), with the value of $1.5 \times 10^{10}$ m$^{-3}$. The mesopause temperature undergoes mild fluctuations, but all changes are well within the errors and thus, this event is classified as stable. The relative OH band intensity strongly increases (from 460 to 1079) during half an hour before the onset. After the event onset a more steady and gentle increase in the emission intensity is seen. A more persistent mesospheric electron density enhancement below 95 km may refer to an evolution of sporadic E layer which can affect the temperature evolution. Furthermore, the short lifetime of the electron density enhancement is a good candidate to explain the temperature stability.

*The seventh event* begins at 02:40 UT on 25 January 2014. It is detected at an altitude range of 87–90 km with an electron density value of $2.6 \times 10^{10}$ m$^{-3}$. The electron density enhancement is strong for about 20 min and remains at an elevated level for the next two hours. The mesopause temperature stays stable over the event onset. The OH band intensity values also keep constant with respect to the intensities an hour before the EPP event.

*The eighth event* is detected at 19:50 on 6 February 2016 in the altitude range of of 87–90 km with an electron density value of $5.2 \times 10^9$ m$^{-3}$. This increase in precipitating flux lasts for 50 minutes. The mesopause temperature decreases by 13 K from the pre-EPP level of 195 K to the post-EPP level of 182 K. Comparing the same time steps the emission intensity decreases from 1164 to 911 in correlation with the temperature.

*The ninth event* commences at 21:50 on 5 January 2019 at the altitude of 91–94 km with an electron density value of $2.2 \times 10^{10}$ m$^{-3}$. The electron density measurements are only available until ten minutes after the onset time, but display a steep increase until then. The temperature increase at the EPP event onset is, however, within the uncertainty and therefore classified as stable. The relative band intensity is 1372 half an hour before the onset, 1201 at the onset time and 1058 half an hour after onset, which is a more systematic decrease than what is seen in the temperature.

*The tenth event* starts at 19:50 UT on 6 January 2019 and is detected in the altitude range of 91–94 km. The electron density at the EPP onset is $1.3 \times 10^{10}$ m$^{-3}$, and the enhancement lasts for about 30 minutes. The mesopause temperature does not decrease when comparing the pre-EPP level to that when the electron density is first enhanced. The relative OH(6–2) band intensity at the event onset is 54 units higher than the level before the event onset time, but stays otherwise constant throughout the analysis window.

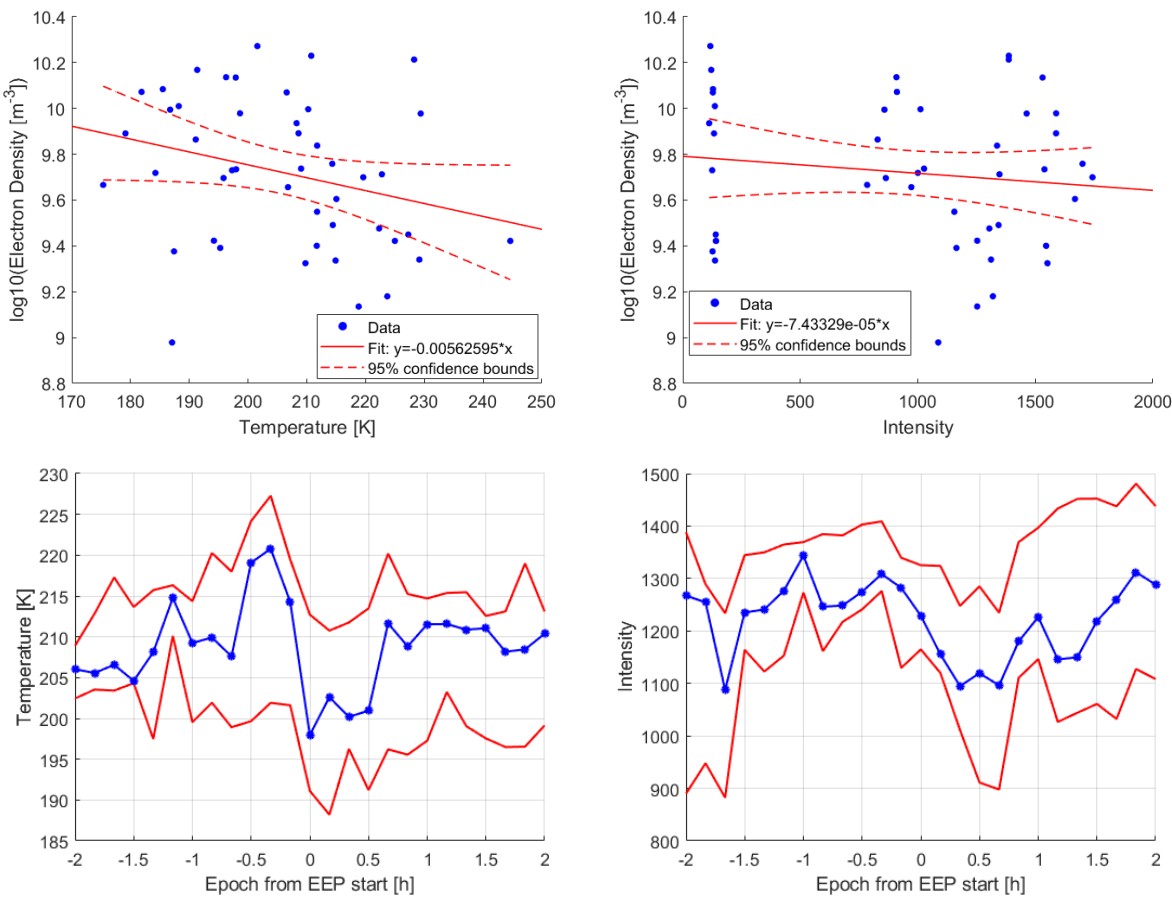

**Figure 3.** This figure illustrates the average temperature and intensity response to the EPP onset. The upper left (right) panel shows a scatter plot of the electron density and the airglow temperature (intensity) values, including data points from half an hour before to one hour after the EPP event, for the six decreasing events. The regression line, with its 95% confidence bounds are shown in red. The superposed epoch of the airglow temperature in the lower left panel (airglow intensity in the lower right panel) includes the 25% (lower red line), 50% (blue) and 75% (upper red line) percentiles of the temperature for all ten events. The *first* and *second event* are accompanied by very low intensity values. To reduce the dominance of these two events the lower percentile for intensity is set at 35%. The zero epoch time corresponds to the EPP onset. The measurements are chosen closest to the respective epoch time and averaged. Each 30 min epoch time bin contains 6–10 temperature (intensity) values. The variation in the number of data points is due to individual missing data points.

The temperature dependence on the electron density for EPP events with temperature decrease is shown as a scatter plot in Figure 3 (top left panel). The scatter plot includes all simultaneous temperature and electron density value pairs from 30 minutes before to 60 minutes after the EPP onset. The result is a mild anti-correlation between the electron density and the airglow temperature with a large range of variation. The correlation coefficient for EPP events with a temperature decrease is -0.2753. The average temporal evolution of the temperature response is seen in the superposed epoch analysis from 2 hours before to 2 hours after the precipitation onset for all ten EPP events (bottom left). The zero epoch time (EPP onset) shows the lowest temperature both in the median (blue), as well as upper and the lower percentile (red) curves. This immediate temperature decrease is of the order of 20 degrees and recovers within half an hour after the EPP onset. Similarly, the scatter plot of electron density and the relative airglow intensity (top right) suggests a mild negative correlation at brightness values around and larger than 1000 counts. The correlation coefficient for EPP events classified as decreasing is -0.1306. The median, upper and lower percentile of the airglow intensity (bottom right) shows about 15% decrease at zero epoch time, which recovers in tandem with the median temperature. The intensities vary in the range from hundreds to thousands.

As it is clear that there are temperature changes before and after the EPP onsets which are of the same order of magnitude as those during the EPP events, we compiled a list of times ("non-events", see Appendix A) when there are no electron density enhancement reaching the heights below 100 km. This set of non-events is equal in number and mimics the diurnal and seasonal distribution of the EPP events (same days). There were not enough data coverage and quiet conditions during the first and second event, and therefore the non-events are chosen on the days of the *third* to *tenth event*. The superposed epoch plot for the non-events is shown in Figure 4. No systematic temperature decrease of the order of tens of degrees at the non-event onset is found. This suggests that it is unlikely to find a median temperature change of over 10 degrees by coincidence, but aligning the EPP onsets emphasizes the mild effect of decreasing temperature at the EPP onset.

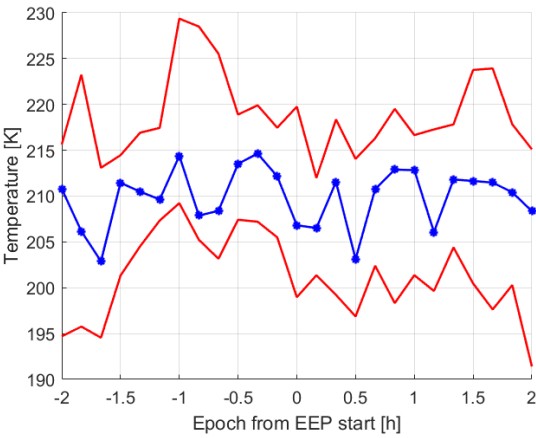

**Figure 4.** Typical temporal evolution of mesopause temperature without EPP onsets. The superposed epoch of the airglow temperature consists of the 25% (lower red line), 50% (blue) and 75% (upper red line) percentiles of the temperature for times with similar background conditions as the EPP events and on the same days as the EPP events (for dates and times see Table A1)

## 5 Discussion

We have found and analyzed ten electron precipitation events which reached the mesopause region and had a good coverage of co-located OH airglow temperature data. Table 1 shows that 6 of 10 EPP events analyzed in this study were accompanied by a decrease in the mesopause temperature by 10–40 K at the EPP onset. The pre-EPP temperature level (temperature values 10–20 minutes before the event) varied between 194 and 229 K, and the temperature decreased to the range of 184–211 K.

The winter mesopause is characterized by a large temperature variability (of the order of 10 K, Suzuki et al. (2010b)) due to gravity wave propagation. This makes it challenging to isolate any changes to be a consequence of any particular driver. The fact that a coherent temperature decrease is seen when the EPP onsets are aligned at zero epoch time does, however, indicate that there is a more systematic response to the particle precipitation, although changes of similar magnitude can occur due to other reasons as well. Sudden stratospheric warmings (SSW), planetary waves, atmospheric tides and the polar vortex strongly affect the mesopause region and its temperature variability (e.g. Harvey et al., 2018), none of these would cause the synchronised temperature changes with EPP onsets. Along with gravity waves these sources can be responsible for temperature changes outside the EPP events. Any of these dynamic events may also act simultaneously with some of the EPP event for enhancing or cancelling the EPP-driven signature, which is likely to be the case with the *first event*, where the temperature decrease is 40 K. Furthermore, the *second* and *tenth event* occur within a week from SSW. Around the *second event* we do not have enough data coverage to see the SSW-related temperature evolution, but in case of the *tenth event* about a week long temperature excursion of the order of 50 K can be seen in the daily averages (not shown) after SSW on 2 January 2019. Despite the slow SSW effect, the temperature around the EPP onset does not change.

It is thus understandable that there is no consensus of the temperature response to EPP when any minor change is likely to be lost in the highly variable background conditions. The temperature responses found in this study for 6 out of 10 EPP events were larger than 10 K. The short lifetime (<60 minutes) of the observed temperature responses does not allow this signature to be detected in analysis utilizing daily averaged temperature data. For instance, the fast temperature decrease and equally quick recovery shown by the superposed epoch analysis (bottom left panel of Figure 3) would not be seen as a change in daily averaged epoch. It may not be significant in the hourly averaged data.

*The first event* (top panel of Figure 2) shows a strong decrease in the mesopause temperature at the time of the EPP onset. Similarly, in *the third event* and *eighth event* quiet ionospheric conditions (low electron density) are seen prior to the EPP onset, and the connection between the electron density enhancements and the temperature changes is clear. This is a typical behavior in our set of events. It is further demonstrated by the anti-correlation between the airglow temperature and the electron density seen in the top panel of Figure 3.

In *the second*, *forth* and *fifth* the short-lasting electron density enhancements are followed by longer lasting but elevated levels of electron densities. In all these cases the electron fluxes reaching the E region heights (above 100 km) are high for about two hours after the EPP event, but only at the very beginning of the precipitation event did they reach the altitudes below 100 km. Consequently, the temperature behaviour is smooth and steady apart from the short-term change around the EPP onset.

*The second event* shows the same characteristics of an apparent decrease. However, the decrease is of the order of the fitting error and thus includes a large uncertainty.

Our *sixth*, *seventh*, *ninth* and *tenth event* were classified as stable in terms of the mesospheric temperature response. The electron density increase in *sixth event* has the shortest lifetime of all our events, as it is seen in one 10-min profile only. Similarly, the *tenth event* is short-lived. The lifetime of the electron bombardment may thus be the key factor determining
whether a measurable neutral temperature response is seen. Based on the events analysed in this study, the temperature change does not directly relate to the precipitating particle flux. More determining factors can be the characteristic energy and the lifetime of the EPP event as well as the ionospheric conditions before the EPP onset. The precipitation properties should be investigated in the future to better understand the determining factors of the EPP impact.

A statistical study on high-latitude OH airglow temperatures and emission intensities by Shepherd et al. (2007) shows a
300 strong positive correlation between the two parameters in the time scales from hours to seasons. This is explained by vertical motion of the airglow layer driven by atmospheric dynamics. For instance, as an airglow layer undergoes downward motion the adiabatic heating increases its temperature. The lower peak emission height coincides with higher mixing ratio of oxygen and therefore, enhances the production of the excited OH. The temperature changes observed in our study take place in shorter time scales. The correlation between the OH(6–2) temperature and the relative band intensity of 30-minute averaged data can
be visually inspected in Figure 2. While a positive correlation can be seen between the two parameters in case of the *third*, *fifth and eighth* event, only mild anti-correlation across the entire event set was found (shown in Figure 3). As the scatter plots include data points from 30 minutes before to 60 minutes after the onset time, the lack of scatter correlation suggests that there is no larger-scale or periodic coherent behaviour between temperature and brightness within the examined time period. The synchronous decrease in temperature and brightness seen in the epoch curves is a short-term feature, which does not dominate
the scatter. A periodic out-of-phase relationship between temperature and brightness, which has been observed for non-EPP conditions Suzuki et al. (2010a) would result in low correlation but would not explain the synchronous decrease at onset.

An increase in the mesospheric temperature during particle precipitation would agree with the Joule heating effect suggested by earlier studies (Nesse Tyssøy et al., 2010). In fact, a temperature increase of about 10 K was observed in the study by Suzuki et al. (2010b). In our study, however, the mesopause temperature responded to the particle impact with a decrease of
315 about 20 K in 6 out of 10 events. A way to explain the temperature change Suzuki et al. (2010b) discussed was that the EPP ionisation changes the mesospheric chemical composition by decreasing the population of excited OH at the top of the layer. As a consequence, the peak height of the airglow changes and the temperatures are weighted by lower altitudes than before. The energetic electron impact can dissociate oxygen and ozone molecules in the mesosphere (e.g. Turunen et al., 2016). When less $O_3$ is available, less excited OH molecules are produced as ozone is a key ingredient in the production of excited hydroxyl:

$$H + O_3 \rightarrow O_2 + OH^*(v' \leq 9), \tag{1}$$

where $v'$ corresponds to the upper vibrational level of the OH molecule, which in our case is 6. The dissociation of molecular oxygen and ozone by energetic electrons can therefore lead to a decrease in the emission of the OH airglow. The fitted rotational

OH temperature corresponds to the height of the airglow layer. The peak is assumed to be at about 87 km. If, however, the production of excited OH is temporarily prohibited at the top part of the airglow layer, the temperature will then represent the layer, which is now centered at lower heights. Depending on the local gradient in the mesospheric temperature profile, this may lead to increased, decreased or unchanged temperature value. In this scenario, the relative OH(6–2) band intensity would decrease as the airglow layer becomes thinner, which is true for most of the events analyzed in this study. In particular at the top of the mesosphere the temperature can vary on the order of 10 K over a height range of a few kilometers. Mesospheric winter time temperatures are also often constant over a large range of heights, which results in no change in the temperature even if the OH layer height changes. This may be the case in our events 6, 7, 9 and 10 where no obvious temperature change was observed.

According to the superposed epoch behaviour in Figure 3, a typical temperature decrease at the event onset is about 20 K, while that for the relative band brightness is about 15%. On a Gaussian airglow profile (as for instance depicted by Suzuki et al. (2010b) in their Figure 4) an intensity reduction of 15% would correspond to thinning of the airglow layer by about 2 km, and a gradient in the temperature profile of about 10 K/km. While no measured mesospheric temperature profiles were available during the events analysed here, browsing polar night temperature measurements by SABER/TIMED spacecraft showed that a decreasing temperature of 5–10 K/km over the airglow altitude range is not uncommon. However, the first event with a temperature decrease of 40 K is not realistically explained by the depletion of OH alone.

The superposed epoch behaviour of non-EPP times which mimic the same diurnal and seasonal distribution as our EPP events (see Figure 4) does not show any transient change of the order of some 10 K at or around the zero epoch time. This suggests that the temperature change in the order of some 10 K is unlikely to appear as a median behaviour by coincidence but is, in the case of this study, a result from aligning the EPP onsets.

Our results together with the previous results by Suzuki et al. (2010b) present an inconsistent temperature response to EPP. Therefore, a larger number of events should be collected and examined to investigate which precipitation parameters and background ionospheric conditions play key roles in the final outcome. Furthermore, an immediate temperature response and its fast recovery suggests that the longer-term and larger-scale heat balance in the mesosphere is little affected by EPP, unless the actual precipitation has a substantial lifetime (hours to days).

## 6 Conclusions

A total of 10220 hours of electron density measurements were browsed in the search of enhancements due to energetic particle precipitation (EPP) events with simultaneous temperature calculations from OH airglow measurements. A total of ten events of electron density enhancements were found and analyzed in this study. Although the number of events is not statistically sound, the results are systematically pointing to a short-term EPP effect on neutral temperature based on co-located measurements and in particular, direct electron precipitation measurements. We searched for any coherent behaviour between the electron density enhancements at the D region heights and the mesopause temperature. The response of the mesopause temperature on the EPP energy deposition is predominately (6 out of 10 events) an immediate decrease of 10–40 K, which recovers within less

than 60 minutes after the EPP onset. In case of 4 events the temperature change was only a few degrees and well within the uncertainty. Our findings together with a temperature increase in a previous study suggests that an EPP ionisation may decrease the production of the excited OH at the top of the airglow layer. As a consequence, the airglow layer becomes thinner, the peak

height is reduced and the airglow temperatures correspond to lower altitudes. Investigating the change in the relative OH(6–2) band intensity shows a decrease during the majority of our EPP events, thus supporting the thinning scenario as a valid mechanism for changing the measured temperature. Furthermore, the relative OH brightness values are only poorly correlated with the temperatures in the time scales of a few hours, which is not in agreement with purely dynamically driven temperature changes. Four events showing no change in temperature may indicate that the mesospheric temprature stays constant over a

larger range of heights. Given the short-lived characteristic of observed atmospheric temperature changes, EPP is unlikely to have climate effects except for long-lasting events.

*Data availability.*    The temperature data are available as quicklooks plots online at kho.unis.no. The EISCAT Svalbard Radar data have been downloaded from https://eiscat.se and are the intellectual property of the EISCAT Scientific Association. The EISCAT raw data files are analyzed by using the Grand Unified Incoherent Scatter Design and Analysis Package (GUISDAP) Lehtinen and Huuskonen (1996).

## Appendix A: Appendix A

| #  | Event Date [UT]     |
|----|---------------------|
| 1  | 2013/12/06 19:00    |
| 2  | 2014/01/21 01:00    |
| 3  | 2014/01/21 20:00    |
| 4  | 2014/01/24 20:00    |
| 5  | 2014/01/25 01:00    |
| 6  | 2014/01/25 16:00    |
| 7  | 2016/02/06 19:00    |
| 8  | 2019/01/05 19:00    |
| 9  | 2019/01/06 11:30    |
| 10 | 2019/01/06 18:00    |

**Table A1.** As a comparison data set some dates for times without an EPP onset are given here. The dates have been chosen with the similar background conditions as for the EPP events (on the same days).

*Author contributions.* Florine Enengl carried out the analysis of the data and the writing of the paper. Noora Partamies proposed the idea of the study, fitted and pre-analyzed the airglow temperature data, shared her expertise and together with Nickolay Ivchenko took part in the discussions, interpretations, planning and structure of the work. Lisa Baddeley advised and helped describing and analysing the EISCAT data. All authors helped in the writing process with comments, suggestions and edits on the paper.

*Competing interests.* No competing interests are present.

*Disclaimer.* The data and figures have been used in the MSc thesis by F. Enengl, available through http://kth.diva-portal.org/

*Acknowledgements.* The work by NP & LB is supported by the Research Council of Norway (NRC) under CoE contract 223252, and NP is further supported by the NRC contract 287427. The authors thank Fred Sigernes and Mikko Syrjäsuo for the maintenance of the OH airglow spectrometer. EISCAT is an international association supported by research organisations in China (CRIRP), Finland (SA), Japan (NIPR and ISEE), Norway (NFR), Sweden (VR), and the United Kingdom (UKRI). The authors thank Ingemar Häggström for his assistance with the EISCAT data.

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
