# Peer review of "On the relationship of energetic particle precipitation and mesopause temperature"

_Annales Geophysicae, 2020_

## Referee Comment (RC1) · Anonymous Referee #1 · 6 Aug 2020

Review of manuscript: angeo-2020-44
On the relationship of energetic particle precipitation and mesopause temperature

Florine Enengl, Noora Partamies, Nickolay Ivchenko, and Lisa Baddeley

1.  The authors set out to investigate whether electron precipitation events lead to changes in the temperature of the mesopause region. They note that there are conflicting reports on this subject, and they set out to resolve the question.

2.  To do so they use electron density values from the EISCAT radar, and temperatures from the mesopause region derived from OH emissions both of which are recorded from Svalbard.

3.  They start off with over 10,000 hours of radar data which sounds very impressive, but when the selection criteria are applied, it turns out that only eight events remain with which to carry out the investigation. The criteria are clearly stated and the reasons for choosing them are also clear. However, the authors must realise the weakness of undertaking a statistical study with so little data.

4.  The next and biggest problem lies in the time resolution of the temperatures. The authors state that the effect of the EPP on temperature (if it exists) is short lived of the order of 30 minutes (page 12, line 217), and they use 30-minute averages of the OH* temperature instead of the more usual 1-hour averages in an attempt to overcome this. Unfortunately, temperature values are missing either immediately before or immediately after five of the eight EPP events selected for study. Why are there so many missing temperature values? Are the OH spectra contaminated by auroral emissions caused by the precipitating electrons?

5.  The authors claim to have detected a decrease in OH temperature greater than 10 K (10 - 50 K) following the onset of an EPP in seven out of eight cases.

6.  The authors classify the *fifth event* as one of decreasing temperature. This is very strange since the temperature decrease occurs before the occurrence of the EPP, while the temperature has increased by 22 K only ~11 minutes (~02:31 UT) after the maximum value of the EPP (~02:20 UT). This is one of the three events in which there are no missing temperature values either immediately before or immediately after the onset of the EPP. This event should be classified as one of increasing temperature. It is also a pity that no OH temperatures are available after 03:10 UT, since the electron density values remain consistently high until at least 05:10 UT.

7.  The mechanism suggested to explain the perceived temperature decrease, originally proposed by Suzuki *et al*. (2010), envisages a depletion in the number of emitting radicals at the upper part of the OH layer, the effect of which depends on the mesopause region temperature profile at that time. The time resolution of the Suzuki *et al*. (2010) report was 1 minute which is in stark contrast with the present study. Suzuki *et al*. (2010) found support for their proposal from OH VER profiles from the SABER instrument onboard the TIMED satellite. As an absolute minimum, the authors of the present manuscript should at least search for SABER temperature and OH VER profiles, or alternatively, Aura MLS temperature profiles close to the time of the eight events to try to support their argument.

8.  A depletion of OH emitters in the upper part of the layer, leading to probing temperatures at a lower altitude could have the effect of increasing or decreasing the temperature depending on the gradient of the temperature at the time of the measurement (lines 258/259). The winter mesopause temperature is indeed quite variable as pointed out in (lines 213/214). On

average (see e.g., MSISE-90) however, the gradient in the high-latitude winter temperatures profile tends to be small, and the altitude of temperature minimum tends to be above the OH layer. In this situation, a depletion in the upper part of the layer would give rise to a small increase in the OH temperature, with a corresponding decrease in the integrated emission signal.

9. However, average conditions may not be a lot of help here. At any given time, the mesopause region temperature profile is rarely at the average value, and since the time scale of the EPP effect is expected to be short, and with only eight events available for this study, it is unlikely that assuming average conditions would lead to the correct prediction. Nevertheless, it would be surprising to find a temperature decrease in all cases. As stated already, the OH temperature data do not have sufficient time resolution to make a convincing case.

10. The mechanism proposed for the temperature change (decrease), namely depleting the OH layer from above by the precipitating electrons, would be unlikely to give rise to the magnitude of the changes claimed (20-50 K). An approximate calculation based on a 10 K/km vertical gradient over the entire width of a typical Gaussian layer (which would be an extreme case) with a total depletion of say 30% would only change the recovered temperature by ~11 K. At most, one might expect only a few K change in temperature one way or the other with the proposed mechanism. The authors should address this question in detail, i.e., how much of a depletion would be required for a given temperature profile to achieve the temperature changes claimed with the mechanism proposed.

11. The ideas contained in the manuscript have merit, but the data presented is insufficient to support the claim. The temperature data does not have the time resolution needed, and more data are needed to support the premise before publication is warranted.

**Minor comments**

P1, line 4; replace "exited" by "excited".
P1, line 15; replace "events" by "event".
P2, lines 30/31; suggest moving "was found" from the end of the sentence to after "40 K".
P2, line 41; is "deepest" the most appropriate word here? Consider "largest" or "strongest.
P2, line 46; to what does "earlier" refer in this sentence. Do you mean previous reports of EPP events? Is it necessary to include "earlier"?
P3, line 71; be consistent in the use of uppercase- or lowercase-R in "EISCAT Svalbard Radar"; see e.g., P1, lines 5 and 6; P14, line 283.
P4, line 93; replace "field of view" by "field-of-view".
P4, line 107; replace "field aligned" by "field-aligned".
P4, lines 115/116; replace "M. S. Lehtinen (1996)" by "(Lehtinen and Huuskonen, 1996)".
P5, line 138; insert "2019" after "January".
P6, Figure 1, upper panel; omit the text "Produced@DESKTOP- ... 2020".
P10, line 166; replace "(18°)" by "(18 K)".

P12, line 213; the sentence beginning "At high latitudes the …" restates information provided already on page 4 in lines 95-97.
P13, line 243; "Figure 3" should be "Figure 2".

---

## Referee Comment (RC2) · Anonymous Referee #2 · 23 Aug 2020

General comments: The paper has attempted to reveal an effect of the energetic particle precipitation (EPP) to polar mesopause region by analyzing large data set obtained from EISCAT radar and collocated OH airglow spectrometer in Longyearbyen. Authors has selected 8 EPP events which reach mesopause region (i.e. OH layer) by setting criteria in a temporal variation of electron density. As a result of a comparison between OH rotational temperature (T_OH) and electron density at mesopause, they show decrease in T_OH during onset of the EPP event in most cases (7 events of 8). They discussed the cause of this decreasing and concluded that EPP eroded upper part of exited OH (OH*) layer and modulated vertical profiles of OH airglow which can change ground observed T_OH since T_OH is an averaged temperature of neutral atmosphere over the OH* layer. This mechanism has been already suggested by Suzuki

et al. [2010] based on the single EPP events observed in Antarctica. Suzuki et al. [2010] shows increase in T_OH during EPP event and inconsistent with results of the present study. However, substantial effect on OH airglow layer by EPP is modulation OH* profile. Thus, it is possible to observe either increase, decrease, or no change of T_OH during EPP since it depends on a vertical gradient of atmospheric temperature over OH layer. Authors insist that present study support previous studies but showing new feature in variation of T_OH during EPP in polar mesopause region (decrease in T_OH during EPP).

The reviewer evaluates and agrees with the objective and motivation of this study. Quality of data set and analysis direction is also fine. However, the reviewer does not think the current version of the manuscript merits the publication because of lack of substantial verifications to their results and analysis. Major concerns for the reviewer are described below.

1. Discussion and further analysis focusing on variation in OH intensity are required. The authors mainly show variation in T_OH before, during, and after EPP. However, they do not show quantitative verification for intensity of OH airglow (I_OH) before, during, and after EPP as well. Since modulation of height profile of OH* airglow is an essential phenomenon to explain the observed T_OH, the authors have to show more detail and quantitative verifications for observed I_OH. For example, relative amplitude of decrease in I_OH during EPP is necessary to be quantitatively addressed. And then the amplitude has to be verified whether it is enough to change the T_OH with observed level. Empirically Modeled or observed background atmospheric temperature profile and typical profile of OH* intensity would be necessary for this verification. For background temperature profile, satellite data (MLS/AURA or SABER/TIMED) are best to be hired. If coincide temperature profile data are difficult to collect on event days, empirical model (e.g. CIRA) is another choice to know the typical background temperature profile. Anyway, the reviewer strongly recommends authors to check the typical temperature gradient during events weather observed decrease in I_OH can reproduce

observed T_OH.

2. Insufficient discussions to explain the observed variations in T_OH. As the authors mentioned in the manuscript, atmospheric parameters are highly variable in polar mesopause mainly due to existence of many kinds of atmospheric waves. In particular, small scale ($\sim$10 − 100 km) atmospheric gravity wave is known to be major source causing large fluctuations with a period of hours to minutes. Authors excluded this possibility since the correlation between observed I_OH and T_OH is poor and amplitude of T_OH is greater than 10 K for all cases. Nevertheless, the authors also say that decrease in I_OH is shown in most cases (L260). In addition, authors also say that 'While a positive correlation can be seen between the two parameters in case of the fourth and fifth event, no significant correlation across the entire event set was found (data not shown).' (L243). Thus, it seems little bit inconsistent in their context explaining a relationship between T_OH and I_OH. Thus, the reviewer recommends the authors to re-organize their discussion about relationship between T_OH and I_OH. As the reviewer already pointed in former comments, authors must show more details for observed I_OH during EPP events. The amplitude over 10 K is possible and often seen in T_OH due to atmospheric gravity waves in a polar mesopause region. The phase between I_OH and T_OH is roughly positive but can shift each other depending on a vertical wavelength, damping factor, and a sign of vertical wavenumber of atmospheric gravity waves [Liu and Swenson, 2003]. Thus, authors should discuss more carefully to evaluate and exclude the effect of atmospheric gravity waves. For example, in a first event (29 Dec, 2007), there seems large fluctuation with period of 2-hours over the night in both T_OH and I_OH. In this case, phase of T_OH seems to lead the I_OH. This kind of feature is very common and typically observed in variation of T_OH even on no EPP days [e.g. Suzuki et al. EPS., 2010]. https://earth-planets-space.springeropen.com/articles/10.5047/eps.2010.07.010

3. Lack of verification on auroral contamination to OH spectrum data. During the night with active EPP, bright aurora feature would covers entire sky in typical. Since Minel

Interactive
comment

[Figure]

OH(6-2) band sits on wavelength between 825 nm and 860 nm, strong contamination from aurora light (including strong OI line at 844.6 nm) can disturb OH spectrum. Since T_OH is very sensitive to relative intensity of P lines, authors should address the how they judge the spectrum data is free from auroral contamination. The authors mentioned about accuracy of T_OH observation in section 2.2 as +/- 2 K. However, data shown in Fig 1 and Fig 3 have much larger error than this. The authors also should clarify about this point.

Minor commnets:

Fig 1. Include a plot of I_OH as well as Fig 3. Fig 1. Include a vertical line to show the onset time in the plot. Table 1. Add uncertainties in each value. L251 Reference Maeda [1967] is old. The reviewer suggests to add a recent paper modeling O3 destruction during EPP events. (e.g. https://agupubs.onlinelibrary.wiley.com/doi/full/10.1002/2016JD025015%4010.1002/%28ISSN%292169-9402.EEL15)

[Figure]

---

## Author Comment (AC1) · 7 Oct 2020

**Response to Referee 1**

We thank Referee 1 for thorough reading of the manuscript and helpful comments. Below are the point-by-point responses (blue) to each of the comments (black). During the revision process a small timing error was found in the analysis of the spectrometer data. Thus, the temperature and brightness time series plots in Figure 1 and 2 have been re-plotted and have changed a little. This update has resulted in another event to be classified as stable, so there are now 6 cases showing a temperature decrease and 2 showing no change in temperature as an immediate response to the particle precipitation onset. The temperature super-posed epoch plot is re-done in Figure 3, leading to no significant change. Figure 3 now also includes a superposed epoch plot for the airglow brightness as well.

1. The authors set out to investigate whether electron precipitation events lead to changes in the temperature of the mesopause region. They note that there are conflicting reports on this subject, and they set out to resolve the question.

2. To do so they use electron density values from the EISCAT radar, and temperatures from the mesopause region derived from OH emissions both of which are recorded from Svalbard.

3. They start off with over 10,000 hours of radar data which sounds very impressive, but when the selection criteria are applied, it turns out that only eight events remain with which to carry out the investigation. The criteria are clearly stated and the reasons for choosing them are also clear. However, the authors must realise the weakness of undertaking a statistical study with so little data.

   We do recognise the weakness. At the same time 8 events with co-located measurements is more than a case study, and it is a more direct comparison than temperatures versus indirect all-sky particle precipitation measures. Our main point is to demonstrate that even if there is a change in the temperature it recovers fast and is not going to have a major effect on the larger scale mesospheric temperature, which is often speculated in case of EPP.

4. The next and biggest problem lies in the time resolution of the temperatures. The authors state that the effect of the EPP on temperature (if it exists) is short lived of the order of 30 minutes (page 12, line 217), and they use 30-minute averages of the OH* temperature instead of the more usual 1-hour averages in an attempt to overcome this. Unfortunately, temperature values are missing either immediately before or immediately after five of the eight EPP events selected for study. Why are there so many missing temperature values? Are the OH spectra contaminated by auroral emissions caused by the precipitating electrons?

   Missing values in temperature data mean that fitting the synthetic spectrum to the measured spectrum has not given reliable results. This can happen for several reasons: cloudiness, clouds together with light pollution (for instance the Moon), technical problems (for instance interference), or strong auroral contamination by oxygen emission at 844.6 nm, which is in the middle of the OH(6–2) spectrum. These thing are discussed in, for instance, Sigernes et al. (2003), which is cited. The main reasons for missing data have now been mentioned in the text. All these factors can vary very fast within the narrow field-of-view of the spectrometer, and thus cause SNR to change from measurement to another. We have used threshold values for the fit covariance and variances to define a good fit, according to previous work (Sigernes et al. 2003 and Holmen et al. 2013, also cited), which cover long-time series and much observer expertise. Admittedly some of the missing values may be due to the auroral precipitation, but the most dominant reason is clouds, which the Eiscat radar does not care about. Shorter integration times than 30 minutes were tested but that resulted in more missing values due to lower SNR. The observed lifetime of the temperature changes is, of course, limited by the temporal resolution of the temperature data, but the main point is that it explains why earlier observations with daily averages have not shown any changes. Observing the shorter-term temperature changes also gives some confidence to conclude that these changes will not add up to climatologically significant contributions.

5. The authors claim to have detected a decrease in OH temperature greater than 10 K (10–50 K) following the onset of an EPP in seven out of eight cases.

Our events have been re-plotted due to an unnecessary time shift found in the plotting script. With this slightly different averaging there are now 6 cases out of 8 showing a decrease in temperature, typically of the order of 10–20 K. The temperature change of 50 K during the first event is an extreme, which cannot be explained by the thinning of the airglow layer alone. This is explained in the new version.

6. The authors classify the fifth event as one of decreasing temperature. This is very strange since the temperature decrease occurs before the occurrence of the EPP, while the temperature has increased by 22 K only ∼11 minutes (∼02:31 UT) after the maximum value of the EPP (∼02:20 UT). This is one of the three events in which there are no missing temperature values either immediately before or immediately after the onset of the EPP. This event should be classified as one of increasing temperature. It is also a pity that no OH temperatures are available after 03:10 UT, since the electron density values remain consistently high until at least 05:10 UT.

This has changed due to the time averaging mistake we found in the plotting script. The minimum temperature happens shortly after the onset. It is also more explicitly explained in the revised version that the comparison for the classification is done between the data point before the EEP onset and the one at the EEP onset, where the onset data point is the time-wise closest to the EEP onset time. As for the unfortunate timings for missing data points we can only agree.

7. The mechanism suggested to explain the perceived temperature decrease, originally proposed by Suzuki et al. (2010), envisages a depletion in the number of emitting radicals at the upper part of the OH layer, the effect of which depends on the mesopause region temperature profile at that time. The time resolution of the Suzuki et al. (2010) report was 1 minute which is in stark contrast with the present study. Suzuki et al. (2010) found support for their proposal from OH VER profiles from the SABER instrument onboard the TIMED satellite. As an absolute minimum, the authors of the present manuscript should at least search for SABER temperature and OH VER profiles, or alternatively, Aura MLS temperature profiles close to the time of the eight events to try to support their argument.

The temporal resolution of the OH data used in Suzuki et al. is indeed much higher than the one employed in the study. However, that paper analysed one event, where absorption of the cosmic radio noise was used as the particle precipitation proxy. While the OH spectrometer field-of-view is narrow, that of a riometer is nearly all-sky. Thus, their comparison only works for events where the energetic precipitation fills the whole sky with a uniform precipitation spectrum, which was probably true for that one event. We have looked for SABER/TIMED profiles for temperatures for these events, and found nothing particularly close to Svalbard. Since there are large variations in the temperature profiles during the winter it is not helpful to compare to a measurement half an hour or several hundreds of kilometers away from the ground-based station.

8. A depletion of OH emitters in the upper part of the layer, leading to probing temperatures at a lower altitude could have the effect of increasing or decreasing the temperature depending on the gradient of the temperature at the time of the measurement (lines 258/259). The winter mesopause temperature is indeed quite variable as pointed out in (lines 213/214). On average (see e.g., MSISE-90) however, the gradient in the high-latitude winter temperatures profile tends to be small, and the altitude of temperature minimum tends to be above the OH layer. In this situation, a depletion in the upper part of the layer would give rise to a small increase in the OH temperature, with a corresponding decrease in the integrated emission signal.

A model temperature profile (MSIS or CIRA, as shown in Suzuki et al.) indeed gives polar night temperature profiles with very small gradients, and on average OH layer is located below the major temperature minimum, based on random SABER profiles in the vicinity of Svalbard. However, random profiles we inspected also showed quite large amplitude variations, with Earthward negative gradients of 5–10 K/km not being unusual at the heights from about 80 to 90 km.

9. However, average conditions may not be a lot of help here. At any given time, the mesopause region temperature profile is rarely at the average value, and since the time scale of the EPP effect is expected to be short, and with only eight events

 available for this study, it is unlikely that assuming average conditions would lead to the correct prediction. Nevertheless, it would be surprising to find a temperature decrease in all cases. As stated already, the OH temperature data do not have sufficient time resolution to make a convincing case.

We agree with the average temperature profile not being very realistic (see the reply to point 8.). With the number of events we have, it is still possible that the randomness of the variable atmosphere gives the same sign for the temperature change for 6 out of 8 events. As mentioned earlier the temporal resolution is a limiting factor but it is more important to make the point that the temperature changes do not have much longer lifetimes than the temporal resolution of the OH data. The changes are seen in 30-min resolution but would be averaged over by 1-h resolution.

10. The mechanism proposed for the temperature change (decrease), namely depleting the OH layer from above by the precipitating electrons, would be unlikely to give rise to the magnitude of the changes claimed (20–50 K). An approximate calculation based on a 10 K/km vertical gradient over the entire width of a typical Gaussian layer (which would be an extreme case) with a total depletion of say 30 % would only change the recovered temperature by $\sim$11 K. At most, one might expect only a few K change in temperature one way or the other with the proposed mechanism. The authors should address this question in detail, i.e., how much of a depletion would be required for a given temperature profile to achieve the temperature changes claimed with the mechanism proposed.

Since there were no measured temperature and OH profiles close to Svalbard during the events analysed in this study, we looked for any winter and nighttime temperature and OH profiles measured over Svalbard by SABER to see if the mechanism we propose is realistic. First of all, the typical OH brightness change for our events is about 20% decrease at the EEP onset as suggested by the superposed epoch of the OH band brightness in the new Figure 3. In the attached sample profile plot from January 2019 (orbit 92668, top panel of Figure 1 of this document) the temperature is the blue curve as a function of height. The red curve is an average of the two OH bands measured by SABER/TIMED at 2.0 and 1.6 $\mu$m, which correspond to OH bands at higher (9–7 and 8–6) and lower (4–2 and 5–3) vibrational level transitions than the ground-based measurement (6–2). Averaging is done to follow the procedure by Suzuki et al. (2010). The OH volume emission is scaled to bring it to the same X axis with the temperature values for illustration purposes. This OH layer peaks at 86 km. An OH depletion of 20% would remove the top 2 km of the layer. Assuming that this reduction brings the weighted average of the measured temperature down about 2 km from the peak height, it would decrease the temperature by 20 K, because the temperature under the OH peak height decreases by about 10 K/km. As the temperature epoch plot of our EEP events suggests (new versions of the epoch plots in Figure 2 of this document), the typical temperature change is -20 K. Although the profile shown here is not measured during our EEP events, it illustrates the height variation in the winter and nighttime mesosphere, which based on viewing of several tens of profiles is not uncommon. In the bottom panel of Figure 1 the Earthward negative temperature gradient at the heights of the OH peak if about 7 K/km. However, the temperature decrease of 50 K during the first event cannot realistically be explained by this mechanism alone.

11. The ideas contained in the manuscript have merit, but the data presented is insufficient to support the claim. The temperature data does not have the time resolution needed, and more data are needed to support the premise before publication is warranted.

The temporal resolution is not high enough to follow the evolution of these events in detail, but the point of the study is to show that the immediate change is so short-lived that a longer averaging would not allow to see any change at all. While a larger number of events will be available in the future, the 8 events studied here show complementary temperature evolution as compared to earlier case studies. Here we further employ co-located data with mesospheric temperatures and direct measurements of electron precipitation.

12. Minor comments

The following comments have been implemented as suggested:

P1, line 4; replace "exited" by "excited".

P1, line 15; replace "events" by "event".

P2, lines 30/31; suggest moving "was found" from the end of the sentence to after "40 K".

P2, line 41; is "deepest" the most appropriate word here? Consider "largest" or "strongest.

P2, line 46; to what does "earlier" refer in this sentence. Do you mean previous reports of EPP events? Is it necessary to include "earlier"?

P3, line 71; be consistent in the use of uppercase- or lowercase-R in "EISCAT Svalbard Radar"; see e.g., P1, lines 5 and 6; P14, line 283.

P4, line 93; replace "field of view" by "field-of-view".

P4, line 107; replace "field aligned" by "field-aligned".

P4, lines 115/116; replace "M. S. Lehtinen (1996)" by "(Lehtinen and Huuskonen, 1996)".

P5, line 138; insert "2019" after "January".

P6, Figure 1, upper panel; omit the text "Produced@DESKTOP- ... 2020".

P10, line 166; replace "(18°)" by "(18 K)".

P12, line 213; the sentence beginning "At high latitudes the ..." restates information provided already on page 4 in lines 95-97.

P13, line 243; "Figure 3" should be "Figure 2".

[Figure]

**Figure 1.** Two examples SABER/TIMED measurements of OH emission rate (red, scaled to temperature axis) and temperature (blue) over Svalbard in January 2019. Top panel: Orbit 92668, bottom panel: orbit 92624.

[Figure]

**Figure 2.** This figure illustrates the average temperature and intensity response to the EPP onset. The upper left panel shows a scatter plot of the electron density and the airglow temperature values (before, at and after the EPP onset). The superposed epoch of the airglow temperature in the lower left panel (airglow intensity, lower right panel) includes the 25% (lower red line), 50% (blue) and 75% (upper red line) percentiles of the temperature (intensity) for all eight events. The zero epoch time corresponds to the EPP onset. Each 30 min epoch time bin contains 3–7 temperature (intensity) values, maximizing around the zero epoch time.

---

## Author Comment (AC2) · 7 Oct 2020

**Response to Referee 2**

We want to thank Referee 2 for careful reading of the manuscript and thoughtful comments. Below are the point-by-point responses (blue) to each of the comments (black). During the revision process a small timing error was found in the analysis of the spectrometer data. Thus, the temperature and brightness time series plots in Figures 1 and 2 have been re-plotted and have changed a little. This update has resulted in another event to be classified as stable, so there are now 6 cases showing a temperature decrease and 2 showing no change in temperature as an immediate response to the particle precipitation onset. The temperature super-posed epoch plot is re-done in Figure 3, leading to no significant change. Figure 3 now also includes a superposed epoch plot for the airglow brightness as well.

1. General comments: The reviewer evaluates and agrees with the objective and motivation of this study. Quality of data set and analysis direction is also fine. However, the reviewer does not think the current version of the manuscript merits the publication because of lack of substantial verifications to their results and analysis.

   We have extended discussion and analysis according to the suggestions, as detailed below.

2. Discussion and further analysis focusing on variation in OH intensity are required. The authors mainly show variation in T_OH before, during, and after EPP. However, they do not show quantitative verification for intensity of OH airglow (I_OH) before, during, and after EPP as well. Since modulation of height profile of OH* airglow is an essential phenomenon to explain the observed T_OH, the authors have to show more detail and quantitative verifications for observed I_OH. For example, relative amplitude of decrease in I_OH during EPP is necessary to be quantitatively addressed. And then the amplitude has to be verified whether it is enough to change the T_OH with observed level. Empirically Modeled or observed background atmospheric temperature profile and typical profile of OH* intensity would be necessary for this verification. For background temperature profile, satellite data (MLS/AURA or SABER/TIMED) are best to be hired. If coincide temperature profile data are difficult to collect on event days, empirical model (e.g. CIRA) is another choice to know the typical background temperature profile. Anyway, the reviewer strongly recommends authors to check the typical temperature gradient during events weather observed decrease in I_OH can reproduce observed T_OH.

   We agree this being a weakness of the study and have been collecting more evidence. As pointed out by Referee 1 and as also shown by Suzuki et al. (2010), the model temperature profiles tend to be constant over the height range of the mesospheric OH. We searched for SABER/TIMED temperature measurements for the events analysed in this study but found nothing particularly close to the ground-based observation sites. Since the temperature is very variable at the heights of interest in the polar night, a large temporal (> 30min) or spatial (hundreds of km) separation between the ground- and space-based measurements is not giving a reliable comparison. Thus, we looked for random polar night temperature profiles over Svalbard in the SABER measurements. We further prepared a superposed epoch for the OH band brightness. It shows a typical decrease of brightness of about 20% at the EEP event onset. An example of an OH layer profile (red) and temperature (blue) over Svalbard in January 2019 is shown in an attached Figure 1. Here, the OH volume emission rate is an average of the two OH bands SABER measures at 2.0 and 1.6 $\mu$m, which corresponds to vibrational transitions above (9–7 and 8–6) and below (5–3 and 4–0) the one measured from the ground (6–2). As OH bands originate from slightly different heights we assume that averaging provides a reasonable estimate. The OH emission rate values are scaled to bring the profile comparable to the temperature values for illustration purposes. The peak of this average layer is at 86 km. A reduction of about 20% in the brightness would deplete about 2 km from the top of the layer. If the weighted average temperature is brought downwards by 2 km, the resulting temperature is about 20 K lower, as the temperature gradient under the peak height is about -10 K/km. This agrees with the typical change suggested by the epoch evolution of the temperature. However, this mechanism alone cannot explain the temperature decrease of 50 K during the first event, which has been pointed out in the discussion. By browsing temperature profiles from SABER measurements during polar night, it seems that an Earthward negative temperature gradient of 5–10 K/km is not uncommon, but a gradient steeper than that is rare. In the bottom panel of Figure 1, as an additional example, the Earthward negative temperature gradient at the heights of the OH peak if about 7 K/km.

3. Insufficient discussions to explain the observed variations in T_OH. As the authors mentioned in the manuscript, atmospheric parameters are highly variable in polar mesopause mainly due to existence of many kinds of atmospheric waves. In particular, small scale (~10–100 km) atmospheric gravity wave is known to be major source causing large fluctuations with a period of hours to minutes. Authors excluded this possibility since the correlation between observed I_OH and T_OH is poor and amplitude of T_OH is greater than 10 K for all cases. Nevertheless, the authors also say that decrease in I_OH is shown in most cases (L260). In addition, authors also say that 'While a positive correlation can be seen between the two parameters in case of the fourth and fifth event, no significant correlation across the entire event set was found (data not shown).' (L243). Thus, it seems little bit inconsistent in their context explaining a relationship between T_OH and I_OH. Thus, the reviewer recommends the authors to re-organize their discussion about relationship between T_OH and I_OH. As the reviewer already pointed in former comments, authors must show more details for observed I_OH during EPP events. The amplitude over 10 K is possible and often seen in T_OH due to atmospheric gravity waves in a polar mesopause region. The phase between I_OH and T_OH is roughly positive but can shift each other depending on a vertical wavelength, damping factor, and a sign of vertical wavenumber of atmospheric gravity waves [Liu and Swenson, 2003]. Thus, authors should discuss more carefully to evaluate and exclude the effect of atmospheric gravity waves. For example, in a first event (29 Dec, 2007), there seems large fluctuation with period of 2-hours over the night in both T_OH and I_OH. In this case, phase of T_OH seems to lead the I_OH. This kind of feature is very common and typically observed in variation of T_OH even on no EPP days [e.g. Suzuki et al. EPS., 2010].

This part was indeed confusing and has been re-phrased to:
"*As the scatter plot includes data points from one hour before to two hours after the onset time, the lack of scatter correlation suggests that there is no longer-term or periodic coherent behaviour between temperature and brightness within the examined time period. The synchronous decrease in temperature and brightness seen in the epoch curves is a short-term feature, which does not dominate the scatter. A periodic out-of-phase relationship between temperature and brightness, which has been observed for non-EPP conditions (Suzuki et al. EPS (2010)) would result in low correlation but would not explain the synchronous decrease at onset.*"
The length of the time series of T_OH and I_OH data of the first event in Figure 2 is not long enough for reliably detecting periods of 2 hours. Even if it may seem that the temperature is leading the brightness variation, they still both show a minimum value at the same time after the particle precipitation onset. Time shifted scatter plots were explored and do not show an improved correlation.

4. Lack of verification on auroral contamination to OH spectrum data. During the night with active EPP, bright aurora feature would covers entire sky in typical. Since Meinel OH(6–2) band sits on wavelength between 825 nm and 860 nm, strong contamination from aurora light (including strong OI line at 844.6 nm) can disturb OH spectrum. Since T_OH is very sensitive to relative intensity of P lines, authors should address the how they judge the spectrum data is free from auroral contamination. The authors mentioned about accuracy of T_OH observation in section 2.2 as ±2 K. However, data shown in Fig 1 and Fig 3 have much larger error than this. The authors also should clarify about this point.

This is a good point. The auroral contamination part has been clarified in the new version: "*An oxygen auroral emission line at 844.6 nm lies in the middle of the OH(6–2) spectrum. The times when that emission intensity overtakes the OH emission intensity (fit covariance greater than 0.5) are excluded in the temperature analysis due an inaccurate fit. Other things causing poor fits and missing temperature values are cloudiness, high background illumination (e.g. scattered moonlight) or technical issues with the instrument. The threshold values for the fit variances have been determined empirically by viewing and fitting large datasets over decades. For consistency we have employed the same threshold values as in the earlier work (Sigernes et al. 2003, Holmen et al. 2013).*"
During the events studied here the auroral activity reached near the zenith where the radar beam was looking along the magnetic field line (about 7 degrees south of zenith), but did not fully cover the zenith-pointing spectrometer field-of-view. The auroral contamination is responsible for some of the missing data points (such as the one close the onset time of event 8), but mostly the optical measurements are collected right next to the energetic precipitation.

*The error values have been further clarified to distinguish between the accuracy of the method itself and the range given in STD: "The accuracy of the method in estimating rotational temperatures is ±2 K. The error bars and uncertainties given for the data in this study represent the standard deviations (STD) over the averaged time, which are typically somewhat larger (see values in Table 1)."*

5. Minor comments:

Fig 1. Include a plot of I_OH as well as Fig 3. Fig 1. Include a vertical line to show the onset time in the plot.

*OH band brightness evolution has been included in Figure 1, and a vertical line showing the onset time as been added.*

Table 1. Add uncertainties in each value.

*Temperature values in Table 1 have been supplemented with ±standard deviation.*

L251 Reference Maeda [1967] is old. The reviewer suggests to add a recent paper modeling O3 destruction during EPP events. (e.g. https://agupubs.onlinelibrary.wiley.com/doi/full/10.1002/2016JD0250154010.1002/28ISSN292169-9402.EEL15)

*The old reference has been replaced by the recommended one.*

[Figure]

**Figure 1.** Two examples SABER/TIMED measurements of OH emission rate (red, scaled to temperature axis) and temperature (blue) over Svalbard in January 2019. Top panel: Orbit 92668, bottom panel: orbit 92624.

---

## Referee Report (RR1)

The authors investigate the potential for energetic particle precipitation events to directly influence the neutral atmospheric temperature profile in the mesopause region, which is probed via OH rotational temperature measurements from a spectrograph on Svalbard. Eight individual EPP events are discussed after sifting through 10,000 hours of radar data with strict, but appropriate selection criteria. A mechanism involving the depletion of the OH layer is suggested to account for the observed temperature change, and a conclusion that the recovery in temperature happens quickly enough to have little effect on the large scale mesospheric temperature is reached.

1. General comments:
   - The reviewer agrees with the motivation behind the study, and is happy with the quality of data and method of analysis. Response to the initial reviews is appropriate and thorough and the adjustments made are sufficient to support the conclusions presented within.
   - Some clarification/rewording needed in L4-5 (abstract) – it's explained later than you use all manda/ipy experiments between the start of the IPY and Feb. 2019, but this sentence reads as if the experiment was run continuously between the IPY and Feb. 2019.
   - In Table 1. perhaps display the quantitative magnitude of the change in temperature as well as the classification of increasing/decreasing/stable
   - During discussion of $3^{rd}$ event (L178 – 183) the author states: 'The minimum temperature is measured at the time of the electron density maximum' however in the corresponding plot of Figure 2 there is no temperature measurement coincident in time with the electron density peak (at ~22:50UT).
   - Some clarification needed during discussion of the superposed epoch analysis (L213-222). Are the OH temperatures/intensities averaged to produce the temperature/intensity values at the EPP onset time (e.g. Epoch 0) the measurements closest to the onset in time (as discussed earlier)? I understand this is implied when generating a superposed epoch analysis, but a statement that the measurements are never exactly aligned with epoch 0 or the EPP onset time would aid clarity.
   - Author states on L219-220 that 'The upper percentile does not show a clear signature of a temperature decrease' whereas the plot shows the upper red curve showing a similar decrease in temperature (at epoch 0) to the median and lower percentile curves. Some clarification needed. L217-218 also states that: 'The zero epoch time (EPP onset) shows the lowest temperature both in the median (blue) and the lower percentile (lower red) curves.' The temperature at zero epoch for the median curve, is not the lowest temperature seen within that curve, so some rewording needed, likely just to clarify that it is a local decrease seen at epoch 0, rather than a global minimum.
   - Author's discussion regarding the potential OH temperature changes driven by large scale atmospheric dynamics (Paragraph containing L248-260) is valid and necessary in the discussion. However, the added lines from L255 onward discuss a scatter plot which is not presented in the article (scatter of OH-I vs OH-T), perhaps include this scatter plot in Figure 3? There is room in the top panel alongside the OH-I vs E-dens. Scatter plot.

2. Minor comments:

Small grammar improvements and some suggestions to re-structure particular sentences (the latter are left to the author's discretion)

- L2 (abstract): Sentence beginning "Recent results, however…" >>> "However, recent results are inconsistent, which leaves the mechanism and effects still unresolved"
- L15 (abstract): "as opposed proxies" >>> "as opposed to proxies"
- L58/59: "the temperature gradient in the mesosphere" >>> "the local temperature in the mesosphere" – clarification, as it's only dependent on the temperature gradient within the OH layer, not the entire mesosphere.
- L60/61: "during the solar cycle 23 and 24" >>> "during solar cycles 23 and 24"
- L77: "the spectrometer" >>> "spectrometer" – no need for 'the' since the spectrometer hasn't been formally introduced yet, just spectrometer measurements is fine
- Figure 1: Caption, last sentence: "The vertical red lines in the mark the time…">>> "The vertical red lines mark the time…" – no need for 'in the'
- L245: "Our *sixth event* and *eight event* was classified…" >> "Our *sixth event* and *eight event* were classified…" – replace 'was' with 'were' since it is now plural (two events)
- L284: Replace 'height' with 'altitudes'

---

## Referee Report (RR2)

**Journal:** ANGEO

**Title:** On the relationship of energetic particle precipitation and mesopause temperature

**Author(s):** Florine Enengl et al.

**MS No.:** angeo-2020-44

**MS type:** Regular paper

**Iteration:** Major Revision

**Special Issue:** Special Issue on the joint 19[th] International EISCAT Symposium and 46[th] Annual European Meeting on Atmospheric Studies by Optical Methods

**Review**

The paper presents a study of temperature changes at the polar winter mesopause during energetic electron precipitation (EEP) associated with geomagnetic storms.  From over 10 years' data between 2007 and 2019, ten events are identified where electron density measured by the EISCAT Svalbard Radar shows distinct enhancements at 80–95 km and overlapping collocated OH airglow observations are available.  Rotational OH temperatures, and intensities, have been derived from analysis of the night-time airglow measurements as 30-minute moving averages.  Six of the 10 EEP events coincide with temperature decreases in the range 10–20 K, with temperatures recovering in less than an hour.  The other four events show no significant temperature change.  The observed temperature changes are interpreted as changes in the vertical profile of the OH airglow layer leading to a larger contribution from altitudes with lower rotational temperatures.

There is considerable scientific interest in understanding the impact of energetic particle precipitation (EPP) on the chemical and physical structure of the Earth's middle and upper atmosphere.  EPP associated with auroral activity, geomagnetic storms, and solar proton events deposit energy into the polar mesosphere and lower thermosphere (MLT), together with Joule heating and the effects of atmospheric waves and tides.  Temperature changes at the mesopause due to natural processes, and long-term trends associated with environmental change, are poorly understood.  This study contributes to ongoing debate about the impacts of space weather and EPP on the MLT region, where heating modifies the thermal structure, chemistry, and circulation with subsequent coupling upwards affecting atmospheric density and satellite drag and downwards drivers linking to climate.  Such observational studies can make an important contribution to understanding the various processes and improving model parametrisations.

The introduction and methodology sections of the paper are reasonably well written with sufficient relevant detail and citing of prior work.  The choice of observational datasets, combined with the analysis techniques, is appropriate for the study.  The main topic area – mesopause temperatures, EPP, and composition – is suitable for the journal.  The results and discussion follow a logical order.  My main concern about the paper is in the discussion of the ten selected events and conclusions about the atmospheric temperature response to EPP.  These concerns are expressed in my two major comments.  I have identified other areas in the text and figures where clarifications are needed and where the presentation and readability could be improved.  My minor comments and suggested edits are listed below the major comments.  In conclusion, I recommend a major revision of the paper, with the authors addressing my main comments and minor points, before the paper is considered further for publication in *ANGEO.*

**Major comments**

1. Temperature changes.  Although the superposed epoch analysis (SEA, Figure 3) shows the lowest temperature (~198 K) coinciding with the time of (superposed) event onset, the ~20 K decrease is from a peak (~221 K) occurring immediately before onset which is not discussed.  Furthermore, the temperature plots for individual events show instances before event onsets where temperature changes of 20 K or more occur in the absence of EPP.  Therefore, I am unconvinced from the results presented that observed temperature changes can be linked to EPP occurrences.  I recommend that the discussion and conclusions are revised to better

describe Figures 2 and 3, and the challenge of distinguishing EPP-temperature correlations from other variability clearly presented. A more rigorous statistical analysis of the temperature variations e.g., SEA of OH airglow temperatures for randomly selected periods *not* associated with EPP, would also help. I also question why, for the events (seventh and ninth) with the highest precipitating electron fluxes at onset, is no discernible temperature change found?

2. Sudden stratospheric warmings (SSWs). Two of the events (#2 and #10) follow within a week of major SSWs (https://csl.noaa.gov/groups/csl8/sswcompendium/majorevents.html) that would have strongly affected the northern hemisphere mesopause region including above Svalbard. Although gravity wave effects on winter mesopause temperatures are considered, the authors should discuss potential effects of these two SSWs as well as more general variability due to planetary waves, atmospheric tides, and the polar vortex that, as recent work shows (e.g., Harvey, V. L., Randall, C. E., Goncharenko, L., Becker, E., & France, J. (2018). On the upward extension of the polar vortices into the mesosphere. *Journal of Geophysical Research: Atmospheres, 123,* 9171– 9191. https://doi.org/10.1029/2018JD028815), extends well into the mesosphere.

Minor comments

**Abstract**

- Line 1. 'Energetic Particle Precipitation' should be 'Energetic particle precipitation'.

- Line 5. 'spectrum of the OH airglow'. The region of the spectrum and OH band system should probably be stated here. The location of the OH spectrometer should also be given in the Abstract.

- Line 5. 'EISCAT' needs to be defined.

- Line 9. 'Most of our 10 electron precipitation events are associated with a temperature decrease of 10–20 K'. The exact number of events showing a temperature decrease needs to be stated. Perhaps 'our' should be changed to 'the selected', or similar- and an indication given of how the 10 events were selected.

- Line 16. 'conclude that this type of particle precipitation event'. What type of particle precipitation event is being referred to here?

- Line 17. 'if the lifetime of the precipitation was much longer than that of a typical EPP event found in this study'. The duration of a typical event in this study needs to be stated.

**1. Introduction**

- Line 21. The definition 'EPP' needs to be added after the first use of 'energetic particle precipitation' here, rather than on line 26.

- Line 27. Define 'NOAA POES'.

- Line 28. Define 'TIMED'. The TIMED instrument needs to be stated.

- Lines 34–35. 'was observed due to precipitation of 250–800 keV protons (an intense solar proton event).' The date(s) of the intense solar proton event should be added.

- Line 26. 'The rotational hydroxyl (OH) airglow temperature' should probably be 'The hydroxyl (OH) airglow rotational temperature'.

- Lines 33–34. 'at 85–90 km, only a minor cooling of 3–4 K'. In what way is 3–4 K only a minor cooling? Does this relate to the SABER instrument measurement accuracy or background quiet-time temperature fluctuations? It should be made clear here, and elsewhere in the paper, what constitutes a significant temperature change at the mesopause region e.g., due to EPP.

- Line 41. Clarify what is meant by 'largest magnetic deflection'.

- Line 43. Clarify what is meant by 'activity'. Geomagnetic activity?

- Line 63. 'The results showed that the mesopause temperature from October to February…' It should be made clear that the results are for the northern hemisphere. Presumably the same might be the case for the southern hemisphere winter months?

- Lines 65–66. 'They concluded on a temperature change of about 4 K per 100 solar flux units (SFU) of the F10.7 radio flux.' Is this temperature change associated with solar UV variability during the solar cycle or EPP? The following sentence suggests that EPP effects are another matter, but earlier in the paragraph solar activity is mentioned in terms of Ap. The effects on the mesopause of changes in solar activity – strongly indicated by F10.7 - and geomagnetic activity - indicated by Ap - need to be clearly distinguished.

**2. Instrumentation**

**2.1 EISCAT Svalbard Radar**

- Line 83–84. 'D-region altitudes.' The D-region isn't yet mentioned. Suggest either include some words about the D-region or change to 'mesopause altitudes'.

**2.2 Ebert-Fastie airglow Spectrometer**

- Line 91. 'rotational OH(6–2) band'. More correctly, the OH(6–2) band is a rovibrational, or rotation-vibration band. However, it is probably correct to say that *rotational* temperatures are derived by analysing the P-branch lines of the band system.

- Line 92. 'at Svalbard latitudes (78.2° N)' should be 'at the Svalbard latitude (78.2° N)'.

- Lines 93–94. 'The spectral resolution of the OH(6–2) band is 0.4 nm.' It needs to be made clear that this is the resolution of the spectrometer measurement. The true linewidth of the OH airglow lines will be much narrower.

- Line 94. 'several scans are averaged'. Please state exactly how many scans are averaged. If half hour averaging is used, I would say that is going to be rather more scans than 'several'.

- Line 95. 'Most earlier studies use 1-hour averages. In this study, half-hour averaging is used'. Presumably 1-hour averages were chosen to reduce signal-to-noise in the airglow spectra to an acceptable level. What is the consequence on signal-to-noise, and temperature determinations, of the shorter, 30 minute averages used in this study?

- Line 99. 'An oxygen auroral emission line at 844.6 nm'. It should be made clear that the auroral emission is from *atomic* oxygen.

- Lines 104–105. 'at the event selection state. Once the events were selected'. It needs to be clarified what are the events being selected?

- Line 109. Clarify what is meant by 'The error bars shown in this study represent the standard deviations (STD) over the averaged time'. How can standard deviations over the averaged time be determined from, as suggested earlier, averaged spectra?

**3. Data description and event selection**

- Line 115. 'These experiments provide a sufficient height resolution'. State what is a sufficient height (altitude) resolution.

- Line 119. 'from 1 March 2007 to 29 February 2008 (Blelly et al., 2010). This year includes 8784 hours in the ipy experiment mode.' Which year is being referred to here?

- Line 121. 'The total of 1388 hours of ESR data between 2008–2019 were analyzed in more detail.' In more detail than what? Also, 'were' refers to the total and therefore should be 'was'.

- Line 126. Perhaps the model atmosphere used could be stated / referenced?

- Line 129. Why were the altitude ranges 87–90 km and 91–94 km chosen? Perhaps the electron densities at lower altitudes (i.e., <87 km), overlapping the lower region of the OH airglow layer, are too noisy?

- Lines 131–132. Suggest change 'based on an earlier study by Cresswell-Moorcock et al. (2013). The onset is found by a sudden increase of the electron density (median value) by a factor of 5 over 5 minutes.' to 'based on an earlier study by Cresswell-Moorcock et al. (2013), in which onsets were found as sudden increases of electron density (median value) by a factor 5 over 5 minutes.' This then makes it clear what was done previously, compared to the following sentences that describe what was done in this study.

- Line 145. 'which where' should be 'which were'.

- Line 147 and Figure 1. 'The electron density during the radar run on 6 January 2019 from 16:00 UT to 22:00 UT is shown in Figure 1'. It should be pointed out here and in the Figure 1 caption that this example is the tenth event (#10) in the Results section. The lower panels of Figure 1 show essentially the same data as the corresponding plots in Figure 2 for that event, but with different (perhaps better) axis ranges. The temperature and intensity panels could be removed from Figure 1 and the text incorporated in the discussion of the tenth event (lines 222–226).

- Line 148. The electron density at the lower part of the ionosphere'. Please state the altitude range that is meant here.

- Lines 148–149. 'electron density … was low (mainly below $10^{10}$ m$^{-3}$) but abruptly increased at the EPP onset time.' Is the low electron density value for quiet-time conditions before EPP increases, and how abruptly does it increase?

- Line 150. What might be causing the large temperature variations (~20 K) in the hours before EPP onset?

- Line 154. 'During this experiment the radar was pointed to zenith, which is aligned with the spectrometer field-of-view.' This sentence is unnecessary since it has already been stated that the radar and spectrometer are collocated with overlapping views.

- Lines 155–156. I cannot see a missing temperature data point in Figure 1. What is the time of the missing measurement or data point?

**4. Results**

- Line 174. 'The electron precipitation lasts for about 30 min.' should be 'The increased precipitating electron flux lasts for about 30 min.' or similar.

- Line 176. 'relative OH(6–2) band intensity is 148 before the event and only slightly diminishes to 129 over the EPP onset'. Rather than terms such as 'only slightly', it would be better in the description of each event to quantify the change e.g., as a percentage decrease / increase.

- Line 184. 'The soft precipitation'. In what way is the precipitation soft? Since it is detected at 87–90 km, the electron precipitation could be described as harder than for events where it is detected higher up at 91–94 km.

- Line 195. 'stays slightly elevated'. State the value of the electron density that is slightly elevated.

- Line 213. Suggest change 'The light precipitation' to 'This modest increase in precipitating flux' or similar.

- Lines 229–235. The results section discusses correlations between electron density and airglow temperature / intensity in Figure 3. Regression lines for the data need to be added to the scatter plots and the correlation values stated.

**5. Discussion**

- Line 245. '10 degrees' should be '10 K'. Rather than 'often larger…', it should be stated for exactly how many events the temperature change was larger than 10 K.

- Lines 247–248. 'the fast temperature decrease and equally quick recovery shown by the superposed epoch analysis (bottom left panel of Figure 3) would not be seen as a change in hourly or daily averaged epoch'. I agree that daily averaged epochs are unlikely to show the temperature decrease, but hourly averages may do so. Suggest the authors either remove the statement about hourly data or substantiate the claim e.g., by showing the 30-minute data smoothed to hourly averages.

**6. Conclusions**

- Line 302. Suggest change 'downward temperature decrease by 5–10 K/km at the airglow altitude' to 'decreasing temperature of 5–10 K/km over the airglow altitude range', or similar.

- Line 310–311. 'Energetic electron precipitation' is defined as 'EPP' whereas that abbreviation was used previously (lines 1 and 26) for 'energetic particle precipitation'.

- Line 317. 'only a few decrease'. I think this should be 'only a few degrees'.

**Figure 1**

- The upper altitude of the electron density panel should be set at 160 km as there are no plotted data higher up.

**Figure 2**

- The colour bars of the electron density panels are missing titles.

- The vertical axis limits of the electron density, temperature, and intensity panels need to be adjusted to more clearly show the plotted data without significant white space, and without clipping the error bars.

**Figure 3 caption**

- 'The lower percentile for intensity (is) 35%.' Why is the lower percentile set at 35% rather than 25% as for temperature?

- 'Each 30 min epoch time bin contains 6–10 temperature (intensity) values.' Why does the number of values in each bin vary between 6 and 10?

---

## Referee Report (RR3)

The authors investigate a potential correlation between D-region electron density enhancements and neutral (OH) temperature changes. A mechanism which proposes a depletion of the OH layer, as a result of energetic particle precipitation effecting the local chemistry at the mesosphere, is invoked to explain the observed temperature changes. The magnitude and direction of the change in neutral temperature is dependent on the local conditions. They conclude that the proposed mechanism can explain their observations. They also conclude that the recovery in temperature is rapid enough, and the duration of events short enough, that EPP driven temperature change likely has little effect on the long-term heat balance of the local atmosphere.

Responses to the majority of review comments are satisfactory. The increased temporal resolution of the spectral measurements is an appropriate adjustment to the method and further supports the conclusions of the paper.

The reviewer continues to agree with the motivation behind the study and remains happy with the quality of data and method of analysis. Some final adjustments, for the sake of clarity, are suggested below.

General comments:

- Some clarification needed in Section 2.2. The author states that 'the errors bars shown in this study represent the standard deviations (STD) over the average time'. Presumably this refers to the STD of the temperature, which is calculated from the variance of the linear fit? The reviewer understands this could be thought to be obvious, but it should be made much clearer where exactly this number (the STD) comes from, especially considering the role it plays later in determining the classification of the events.
- In Section 4, L:158-159 and Table 1: The author states: 'The criterion for a changing mesopause temperature is that the change has to be larger than the standard deviation of the temperatures averaged over half an hour'. The reviewer believes the STD in question, which is used for the criteria discussed here, is that on the T+2 measurement. This should be made clear. Furthermore, if this is the case, it appears to me that Event 2 does not meet the criteria set out in the text, since it records a delta-T of -13K, and a STD on both T-1 and T+2 of ±21 K. This is briefly addressed in lines 263-264 but isn't concluded on.
- In the discussion of the first event (L174-177), to help with clarity, it should be made clear that this delta T refers to a longer time interval compared to the others, due to the missing temperature measurement at T-1.
- Figure 3, top right scatter plot. The reviewer is glad of its inclusion, although it raises a further question. A number of points all seem to share very similar relative intensities (approximately 200), but this is not commented on. An investigate and discussion of the cause of this feature is needed before their inclusion in the analysis.
- Figure 3 caption: A short sentence reads: 'The lower percentile for intensity 35%'. Please clarify this statement.
- Section 5: Lines 257-258. The author states an anti-correlation is seen between the airglow temperature and electron density. This is correct, but the correlation is mild at best. This is later referred to as evidence against periodic behaviour on the timescale of the observations. The reviewer believes that the inclusion of a subset of points on this scatter, corresponding to the 6 events that show a decrease in OH temperature with increased electron densities, could better highlight a supporting

correlation. Furthermore, the correlations in both scatter plots could be statistically quantified to aid discussion.

Section 5: Lines 299-301 contain a statement that the mesospheric temperature profile can also be relatively steady over the extent of the OH layer, which can result in little to no change in temperature in response to the peak altitude of emission changing. The reviewer believes the intention is to propose that this could be the case during some of the events studied (wherein no significant delta T was observed), as either an alternative explanation to the short lifetimes discussed in lines 267-268, or in addition to. If so, this would be helped by adding another sentence to clarify this.

Minor comments:

• L164: change 'event' to 'events' since it is now plural.

---

## Referee Report (RR4)

**Journal:** ANGEO

**Title:** On the relationship of energetic particle precipitation and mesopause temperature

**Author(s):** Florine Enengl et al.

**MS No.:** angeo-2020-44 (version 5, submitted on 30 Jun 2020)

**MS type:** Regular paper

**Iteration:** Revision

**Special Issue:** Special Issue on the joint 19th International EISCAT Symposium and 46th Annual European Meeting on Atmospheric Studies by Optical Methods

**Review**

In their revised manuscript (version 5) and authors' response (version 3), the authors have satisfactorily addressed my two major comments and most of my minor comments. The presentation of the results is now considerably improved, and the discussion and conclusions sections are more convincing. I have several minor comments which either restate previous ones that have not been completely addressed or are in connection with changes to the text or the new plots that have been added. Responding to these minor comments should be straightforward. In conclusion, I commend the authors (and Editor) for their efforts and perseverance reworking the manuscript and recommend minor revision of the paper before the paper is considered for publication in *ANGEO.*

**Minor comments**

- Lines 28–29. The instrument on the TIMED satellite that provides neutral temperatures should probably be stated.

- Line 43. My previous comment: Clarify what is meant by 'largest magnetic deflection'. The authors' response states Instead of "largest magnetic deflection" we use "lowest local minimum of the horizontal H component", which is more accurate. However, this change has been omitted in the revised manuscript.

- Line 45. My previous comment: Clarify what is meant by 'activity'. Geomagnetic activity? The authors' response states Yes, this refers to geomagnetic activity, and it has been stated now. However, this change has been omitted in the revised manuscript.

- Lines 121–122. 'These experiments provide a sufficient height resolution to detect enhanced electron densities in the mesopause region (<5 km).' The '(<5 km)' should be placed immediately after 'height resolution' as that is what it refers to.

- Figure 3. The units of the electron density labels on both upper plots are unclear (formatting error?) and need to be corrected. Also, the regression line and confidence bounds are faint and need to be made clearer using thicker lines.

- Figure 3 caption. 'confidence bounce' should be 'confidence bounds'. The sentence 'To exclude the dominance of a these two events we increased the lower percentile to 35%.' would be better written as 'To reduce the dominance of these two events the lower percentile for intensity is set at 35%.', or similar wording.

- Figure 4 and Appendix A / Table A1. The selection of non-EPP dates needs to be explained better. The ten non-EPP epochs are on the same dates (but different times) as eight of the ten EPP events (Table 1 – Event #3–10). Why are non-EPP dates for the first and second EPP events (i.e., 2007/12/29 and 2008/02/28) not used?

---

## Author Response (AR2)

[revised manuscript text omitted]

**Response to Referee 3**

We want to thank Referee 3 for careful reading of the manuscript and thoughtful comments. Below are the point-by-point responses (blue) to each of the comments (black). The spectrometer data have been re-analyzed. The integration time remains 30 minutes, but the temperature points are generated every ten minutes. This allows us to see a more continuous (albeit smoothed) behaviour of the temperature. Nevertheless, the main results and key conclusions remain unchanged. Three more events could be included by further relaxing the data selections criteria, which is reflected by larger uncertainties displayed as error bars. One previously included had to be excluded due to too large uncertainties in the calculated temperture values. The re-analyzed events remained in the same temperature-change categories as before.

1. Some clarification/rewording needed in L4–5 (abstract) — it is explained later than you use all manda/ipy experiments between the start of the IPY and Feb 2019, but this sentence reads as if the experiment was run continuously between the IPY and Feb 2019.
   The revised abstract says that we use all the available experiments from the IPY until February 2019.

2. In Table 1 perhaps display the quantitative magnitude of the change in temperature as well as the classification of increasing/decreasing/stable.
   The table has been updated accordingly.

3. During discussion of 3rd event (L178-–183) the author states: 'The minimum temperature is measured at the time of the electron density maximum' however in the corresponding plot of Figure 2 there is no temperature measurement coincident in time with the electron density peak (at ∼22:50UT).
   The event has now been excluded from the study as the uncertainties became too large.

4. Some clarification needed during discussion of the superposed epoch analysis (L213– 222). Are the OH temperatures/intensities averaged to produce the temperature/intensity values at the EPP onset time (e.g. Epoch 0) the measurements closest to the onset in time (as discussed earlier)? I understand this is implied when generating a superposed epoch analysis, but a statement that the measurements are never exactly aligned with epoch 0 or the EPP onset time would aid clarity.
   The data averaging has now been changed so that a temperature point is generated every ten minutes, although we still integrate for 30 minutes. The value at Epoch 0 thus includes data collected during the previous half an hour. The averaging is clarified accordingly in the text.

5. Author states on L219–220 that 'The upper percentile does not show a clear signature of a temperature decrease' whereas the plot shows the upper red curve showing a similar decrease in temperature (at epoch 0) to the median and lower percentile curves. Some clarification needed. L217–218 also states that: 'The zero epoch time (EPP onset) shows the lowest temperature both in the median (blue) and the lower percentile (lower red) curves.' The temperature at zero epoch for the median curve, is not the lowest temperature seen within that curve, so some rewording needed, likely just to clarify that it is a local decrease seen at epoch 0, rather than a global minimum.
   The plots have all been updated due to the re-analysis of the data and wording has been updated accordingly. Both percentiles show clear decreases around the onset times.

6. Author's discussion regarding the potential OH temperature changes driven by large scale atmospheric dynamics (Paragraph containing L248–260) is valid and necessary in the discussion. However, the added lines from L255 onward discuss a scatter plot which is not presented in the article (scatter of OH-I vs OH-T), perhaps include this scatter plot in Figure 3? There is room in the top panel alongside the OH-I vs E-dens. scatter plot.
   Good point. The scatter plot of airglow band brightness and electron density has been included in the new version of Figure 3, and for large intensity values it also shows a mild negative correlation.

7. Minor comments

The following comments have been implemented as suggested:

45    L2 (abstract): Sentence beginning "Recent results, however. . ." » "However, recent results are inconsistent, which leaves the mechanism and effects still unresolved"

L15 (abstract): "as opposed proxies" » "as opposed to proxies"

L58/59: "the temperature gradient in the mesosphere" » "the local temperature in the mesosphere" – clarification, as it is only dependent on the temperature gradient within the OH layer, not the entire mesosphere.

50    L60/61: "during the solar cycle 23 and 24" » "during solar cycles 23 and 24"

L77: "the spectrometer" » "spectrometer" – no need for 'the' since the spectrometer has not been formally introduced yet, just spectrometer measurements is fine

Figure 1: Caption, last sentence: "The vertical red lines in the mark the time. . ." » "The vertical red lines mark the time. . ." – no need for 'in the'

55    L245: "Our sixth event and eight event was classified. . ." » "Our sixth event and eight event were classified. . ." – replace 'was' with 'were' since it is now plural (two events)

L284: Replace 'height' with 'altitudes'

**Response to Referee 4**

We thank Referee 4 for reading and evaluating the manuscript. Below is our response (blue) to the presented comment (*black*).

*I feel that the measurements need to be of much better quality. A photometer using multiple filters to measure the rotational temperature from the ratio of two OH emissions would produce results with much improved precision.*

The instrumentation we have for the airglow is the spectrometer. It measures the OH spectrum, which is then used to fit the intensities of four emission line pairs to give the mesopause temperature. The method itself is robust, although there are other caveats in the data, as discussed in the paper. Comparing temperature values obtained from two different OH bands would be trickier, because they then originate from slightly different altitudes, which would impose further uncertainties in the interpretation.

So instead, the spectrometer data have been re-analyzed. The integration time remains 30 minutes, which is necessary to obtain good signal-to-noise ratio, but the temperature values are generated every ten minutes. This provides a more continuous (albeit smoother) behaviour of the temperature. Nevertheless, the main results and key conclusions remain unchanged, and the re-analyzed events remained in the same categories as before.

---

## Author Response (AR3)

[revised manuscript text omitted]

**Response to Referee 3**

We thank Referee 3 for reading and evaluating the manuscript. Below are our responses (blue) to the presented comments (black).

5     – Some clarification needed in Section 2.2. The author states that 'the errors bars shown in this study represent the standard deviations (STD) over the average time'. Presumably this refers to the STD of the temperature, which is calculated from the variance of the linear fit? The reviewer understands this could be thought to be obvious, but it should be made much clearer where exactly this number (the STD) comes from, especially considering the role it plays later in determining the classification of the events.

10     This is now changed to say that for the averaged spectra the temperature error corresponds to the Boltzmann plot least square fit error. Instead of STD the text uses the word uncertainty or error.

    – In Section 4, L:158–159 and Table 1: The author states: 'The criterion for a changing mesopause temperature is that the change has to be larger than the standard deviation of the temperatures averaged over half an hour'. The reviewer believes the STD in question, which is used for the criteria discussed here, is that on the T+2 measurement. This should

15     be made clear. Furthermore, if this is the case, it appears to me that Event 2 does not meet the criteria set out in the text, since it records a delta-T of -13K, and a STD on both T-1 and T+2 of ±21 K. This is briefly addressed in lines 263–264 but isn't concluded on.

    It is now mentioned that the change shall be larger than the uncertainty and as an exception we additionally include the Second event because dT is large (>10K in accordance with Suzuki et al. (2010b))

20     – In the discussion of the first event (L174–177), to help with clarity, it should be made clear that this delta T refers to a longer time interval compared to the others, due to the missing temperature measurement at T-1.

    The following sentence was added: 'Here the T-1 temperature point is missing, therefore the earliest available point, with its integration time center 10 minutes earlier, is used.'

    – Figure 3, top right scatter plot. The reviewer is glad of its inclusion, although it raises a further question. A number of

25     points all seem to share very similar relative intensities (approximately 200), but this is not commented on. An investigate and discussion of the cause of this feature is needed before their inclusion in the analysis.

    The intensities vary in magnitude from hundreds to thousands. This groups the intensity values roughly into two. This is mentioned in the revised text.

    – Figure 3 caption: A short sentence reads: 'The lower percentile for intensity 35%'. Please clarify this statement.

30     The sentence was corrected and an explanation is given in the text. "The standard" 25% percentile is dominated by two events with extremely low intensity values (100–200) which does not describe the bulk behaviour.

    – Section 5: Lines 257–258. The author states an anti-correlation is seen between the airglow temperature and electron density. This is correct, but the correlation is mild at best. This is later referred to as evidence against periodic behaviour on the timescale of the observations. The reviewer believes that the inclusion of a subset of points on this scatter,

35     corresponding to the 6 events that show a decrease in OH temperature with increased electron densities, could better highlight a supporting correlation. Furthermore, the correlations in both scatter plots could be statistically quantified to aid discussion.

    The new scatter plots include a regression line with 95 % confidence bounds. The correlation values are stated in the text. The scatter plots include only data of the 6 decreasing EPP events in the updated version. The anti-correlation is

40     still mild but it is not a positive correlation.

    – Section 5: Lines 299–301 contain a statement that the mesospheric temperature profile can also be relatively steady over the extent of the OH layer, which can result in little to no change in temperature in response to the peak altitude of emission changing. The reviewer believes the intention is to propose that this could be the case during some of the events

45 studied (wherein no significant delta T was observed), as either an alternative explanation to the short lifetimes discussed in lines 267–268, or in addition to. If so, this would be helped by adding another sentence to clarify this.

text added: This may be the case in our events 6, 7, 9 and 10 where no obvious temperature change was observed.

– Minor comments: L164: change 'event' to 'events' since it is now plural.
In the text it says: "Apart from the ninth and tenth event", which should be correct as singular (on line 177 in the new version).

**Response to Referee 5**

We thank Referee 5 for reading and evaluating the manuscript. Below are our responses (blue) to each presented comment (black).

1. Major comments: 1. Temperature changes. Although the superposed epoch analysis (SEA, Figure 3) shows the lowest temperature (∼198 K) coinciding with the time of (superposed) event onset, the ∼20 K decrease is from a peak (∼221 K) occurring immediately before onset which is not discussed. Furthermore, the temperature plots for individual events show instances before event onsets where temperature changes of 20 K or more occur in the absence of EPP. Therefore, I am unconvinced from the results presented that observed temperature changes can be linked to EPP occurrences. I recommend that the discussion and conclusions are revised to better describe Figures 2 and 3, and the challenge of distinguishing EPP-temperature correlations from other variability clearly presented. A more rigorous statistical analysis of the temperature variations e.g., SEA of OH airglow temperatures for randomly selected periods not associated with EPP, would also help. I also question why, for the events (seventh and ninth) with the highest precipitating electron fluxes at onset, is no discernible temperature change found?

As it is discussed, the winter mesospheric temperature undergoes very large variability in general, which makes it indeed challenging to isolate any changes as a consequence of any particular driver. The fact that a coherent temperature decrease is seen when the EPP onsets are aligned at zero epoch time does, however, indicate that there is a more systematic response to the particle precipitation, although changes of similar magnitude can occur due to other reasons at other times as well. We additionally performed a superposed epoch analysis of randomly selected "non-events", for which EISCAT data show no particle precipitation down to the D region heights. These events were further chosen to mimic the diurnal and seasonal distribution of our EPP events. The resulting temperature epoch (in a new Figure 4) does not show any transient change of the order of some 10 K at or around the zero epoch time. This suggests that the temperature change in the order of some 10 K is unlikely to appear as a median behaviour by coincidence but is, in the case of this study, a result from aligning the EPP onsets. Based on the EPP events analysed in this study, the temperature change does not directly relate to the precipitation electron flux. Other determining factors can be the characteristic energy and the lifetime of the EPP event as well as the ionospheric conditions before the EPP onset. These are open questions for future investigations as they require a much larger event set to allow binning according to different parameters.

2. Sudden stratospheric warmings (SSWs). Two of the events (2 and 10) follow within a week of major SSWs (https://csl.noaa.gov/groups/csl8/sswcompendium/majorevents.html) that would have strongly affected the northern hemisphere mesopause region including above Svalbard. Although gravity wave effects on winter mesopause temperatures are considered, the authors should discuss potential effects of these two SSWs as well as more general variability due to planetary waves, atmospheric tides, and the polar vortex that, as recent work shows (e.g., Harvey, V. L., Randall, C. E., Goncharenko, L., Becker, E., France, J. (2018). On the upward extension of the polar vortices into the mesosphere. Journal of Geophysical Research: Atmospheres, 123, 9171–9191. https://doi.org/10.1029/2018JD028815), extends well into the mesosphere.

While it is very true that SSWs, PW, tides and the polar vortex strongly affect the mesopause region and its temperature variability, none of these would cause the synchronised temperature changes with EPP onsets. In the revised version we mention these different variability sources along with the earlier discussed gravity waves as being responsible for the temperature changes outside the EPP events. Any of these dynamic events may also act simultaneously with some of the EPP event for enhancing or cancelling the EPP-driven signature, which is likely to be the case with the First event, where the temperature decrease is 40 degrees. Around the Second event we do not have enough data coverage to see the SSW effect, but in case of the Tenth event about a week long temperature excursion of the order of 50 degrees can be seen in the daily averages (SSW on 2 Jan). The possibility of these large scale changes masking the small EPP effect is mentioned in the new version of the discussion.

45    2.  Minor comments:

      Line 1. 'Energetic Particle Precipitation' should be 'Energetic particle precipitation'.
      Implemented
      Line 5. 'spectrum of the OH airglow'. The region of the spectrum and OH band system should probably be stated here.
50    The location of the OH spectrometer should also be given in the Abstract.
      The spectral region is now referred to as "infrared". The specific wavelength region for the OH(6–2) band and the station
      location coordinates can be found in the instrumentation section 2.2.

      Line 5. 'EISCAT' needs to be defined.
55    This abbreviation has been defined
      Line 9. 'Most of our 10 electron precipitation events are associated with a temperature decrease of 10–20 K'. The exact
      number of events showing a temperature decrease needs to be stated. Perhaps 'our' should be changed to 'the selected',
      or similar- and an indication given of how the 10 events were selected.
      This has been changed to: "6 out of 10 analysed events"
60    Line 16. 'conclude that this type of particle precipitation event'. What type of particle precipitation event is being referred
      to here?
      EPP events in the mesopause region added
      Line 17. 'if the lifetime of the precipitation was much longer than that of a typical EPP event found in this study'. The
      duration of a typical event in this study needs to be stated.
65    The lifetime of 30–60 min is given in the text now.

   1.  Introduction
      Line 21. The definition 'EPP' needs to be added after the first use of 'energetic particle precipitation' here, rather than
      on line 26.
      The definition has been added
70    Line 27. Define 'NOAA POES'.
      Defined
      Line 28. Define 'TIMED'. The TIMED instrument needs to be stated.
      Defined
      Lines 34-–35. 'was observed due to precipitation of 250–800 keV protons (an intense solar proton event).' The date(s)
75    of the intense solar proton event should be added.
      The time period of the intense solar proton event has been added.
      Line 26. 'The rotational hydroxyl (OH) airglow temperature' should probably be 'The hydroxyl (OH) airglow rotational
      temperature'.
      Correct. This has been fixed accordingly.
80    Lines 33–34. 'at 85–90 km, only a minor cooling of 3–4 K'. In what way is 3–4 K only a minor cooling? Does this relate
      to the SABER instrument measurement accuracy or background quiet-time temperature fluctuations? It should be made
      clear here, and elsewhere in the paper, what constitutes a significant temperature change at the mesopause region e.g.,
      due to EPP.
      This has been explained now on lines 143–145 as less than 10 K being minor and more than 10 K being major, which
85    follows the description by Suzuki et al. (2010b).

      Line 41. Clarify what is meant by 'largest magnetic deflection'.
      Instead of "largest magnetic deflection" we use "lowest local minimum of the horizontal H component", which is more
      accurate.
90    Line 43. Clarify what is meant by 'activity'. Geomagnetic activity?
      Yes, this refers to geomagnetic activity, and it has been stated now.

Line 63. 'The results showed that the mesopause temperature from October to February...' It should be made clear that the results are for the northern hemisphere. Presumably the same might be the case for the southern hemisphere winter months?

It has been specified that our measurements are from the northern hemisphere.

Lines 65–66. 'They concluded on a temperature change of about 4 K per 100 solar flux units (SFU) of the F10.7 radio flux.' Is this temperature change associated with solar UV variability during the solar cycle or EPP? The following sentence suggests that EPP effects are another matter, but earlier in the paragraph solar activity is mentioned in terms of Ap. The effects on the mesopause of changes in solar activity – strongly indicated by F10.7 - and geomagnetic activity - indicated by Ap - need to be clearly distinguished.

This is related to the temperature change during a solar cycle, so a long-term change driven by solar activity, for which F10.7 cm radio flux is a good proxy. While Ap describes the geomagnetic activity and EPP is one signature of that, there may not be a 1-to-1 correlation between the two. Thus, it is impossible to untangle the different drivers. It is here just stated here that the correlation was found between the long-term measurements of mesospheric temperature and the F10.7 flux.

2. Instrumentation

2.1 EISCAT Svalbard Radar

Line 83–84. 'D-region altitudes.' The D-region isn't yet mentioned. Suggest either include some words about the D-region or change to 'mesopause altitudes'.

Changed to mesopause altitudes

2.2 Ebert-Fastie airglow Spectrometer

Line 91. 'rotational OH(6–2) band'. More correctly, the OH(6–2) band is a rovibrational, or rotation-vibration band. However, it is probably correct to say that rotational temperatures are derived by analysing the P-branch lines of the band system.

This has been changed to rotation-vibration band.

Line 92. 'at Svalbard latitudes (78.2°N)' should be 'at the Svalbard latitude (78.2°N)'.

Changed as suggested.

Lines 93–94. 'The spectral resolution of the OH(6–2) band is 0.4 nm.' It needs to be made clear that this is the resolution of the spectrometer measurement. The true linewidth of the OH airglow lines will be much narrower.

True, this has been clarified

Line 94. 'several scans are averaged'. Please state exactly how many scans are averaged. If half hour averaging is used, I would say that is going to be rather more scans than 'several'.

The number of scans (72 for 30 minutes and 144 scans for 1 h ) has been stated in the text.

Line 95. 'Most earlier studies use 1-hour averages. In this study, half-hour averaging is used'. Presumably 1-hour averages were chosen to reduce signal-to-noise in the airglow spectra to an acceptable level. What is the consequence on signal-to-noise, and temperature determinations, of the shorter, 30 minute averages used in this study?

Improving the temporal resolution reduces SNR, which is reflected by larger temperature errors. For the events analysed in this study, this does not affect the temporal coverage of the data, i.e. the selection criteria for "good" data still applies.

Line 99. 'An oxygen auroral emission line at 844.6 nm'. It should be made clear that the auroral emission is from atomic oxygen.

This has been clarified.

Lines 104–105. 'at the event selection state. Once the events were selected'. It needs to be clarified what are the events being selected?

This has been changed to say that it was the selection of EPP events.

Line 109. Clarify what is meant by 'The error bars shown in this study represent the standard deviations (STD) over the averaged time'. How can standard deviations over the averaged time be determined from, as suggested earlier, averaged spectra?

This is a mistake, thank you for pointing it out! For the averaged spectra the temperature error corresponds to the Boltzmann plot least square fit error, and only daily temperature values will have a STD uncertainty based on the hourly integrated spectra. We call this a temperature error instead.

3. Data description and event selection

Line 115. 'These experiments provide a sufficient height resolution'. State what is a sufficient height (altitude) resolution.

The "sufficient" height resolution has now been described as being less than 5 km.

Line 119. 'from 1 March 2007 to 29 February 2008 (Blelly et al., 2010). This year includes 8784 hours in the ipy experiment mode.' Which year is being referred to here?

This refers to the IPY year from March 2007 until February 2008 and is clarified as "this IPY year"

Line 121. 'The total of 1388 hours of ESR data between 2008–2019 were analyzed in more detail.' In more detail than what? Also, 'were' refers to the total and therefore should be 'was'.

This was changed to: "was analyzed to search for EPP events"

Line 126. Perhaps the model atmosphere used could be stated / referenced?

We removed the note about ionospheric model because it is going very deep into the EISCAT data analysis.

Line 129. Why were the altitude ranges 87–90 km and 91–94 km chosen? Perhaps the electron densities at lower altitudes (i.e., <87 km), overlapping the lower region of the OH airglow layer, are too noisy?

Yes, the electron density measurements become too noisy towards the lower part of the airglow layer, so we chose to only include the heights overlapping with the top part of the airglow layer. This is mentioned in the revised version.

Lines 131–132. Suggest change 'based on an earlier study by Cresswell-Moorcock et al. (2013). The onset is found by a sudden increase of the electron density (median value) by a factor of 5 over 5 minutes.' to 'based on an earlier study by Cresswell-Moorcock et al. (2013), in which onsets were found as sudden increases of electron density (median value) by a factor 5 over 5 minutes.' This then makes it clear what was done previously, compared to the following sentences that describe what was done in this study.

Implemented accordingly

Line 145. 'which where' should be 'which were'.

Changed

Line 147 and Figure 1. 'The electron density during the radar run on 6 January 2019 from 16:00 UT to 22:00 UT is shown in Figure 1'. It should be pointed out here and in the Figure 1 caption that this example is the tenth event (10) in the Results section. The lower panels of Figure 1 show essentially the same data as the corresponding plots in Figure 2 for that event, but with different (perhaps better) axis ranges. The temperature and intensity panels could be removed from Figure 1 and the text incorporated in the discussion of the tenth event (lines 222–226).

Yes, Figure 2 is so compact that we want to show one example event with better axes ranges.

Line 148. The electron density at the lower part of the ionosphere'. Please state the altitude range that is meant here.

The lower part of the ionosphere has been defined as latitudes below 100 km.

Lines 148–149. 'electron density ... was low (mainly below $10^{10}$ m$^{-3}$) but abruptly increased at the EPP onset time.' Is the low electron density value for quiet-time conditions before EPP increases, and how abruptly does it increase?

The low density value refers to pre-EPP value, which is now also referred to as the background. As for the abrupt increase the text has been changed to say: "from the pre-EPP value to a local maximum in 10 minutes" to be more precise.

Line 150. What might be causing the large temperature variations ($\sim$20 K) in the hours before EPP onset?

It could be wave activity as the temperature and the band brightness change in tandem, but this is speculation, and something we cannot determine without additional observations and analysis.

Line 154. 'During this experiment the radar was pointed to zenith, which is aligned with the spectrometer field-of-view.' This sentence is unnecessary since it has already been stated that the radar and spectrometer are collocated with overlapping views.

The latter part of the sentence has been removed. The part stating the radar pointing is necessary because that varies between vertical and field-aligned.

Lines 155–156. I cannot see a missing temperature data point in Figure 1. What is the time of the missing measurement or data point?

This is a mistake, there are no missing data points in this figure. The sentence has been removed.

4. Results

Line 174. 'The electron precipitation lasts for about 30 min.' should be 'The increased precipitating electron flux lasts for about 30 min.' or similar.

The sentence has been changed as suggested.

Line 176. 'relative OH(6–2) band intensity is 148 before the event and only slightly diminishes to 129 over the EPP onset'. Rather than terms such as 'only slightly', it would be better in the description of each event to quantify the change e.g., as a percentage decrease / increase.

Percentages may give a wrong impression when the absolute numbers are small, so we rather just keep the actual numbers. "only slightly" has been removed.

Line 184. 'The soft precipitation'. In what way is the precipitation soft? Since it is detected at 87–90 km, the electron precipitation could be described as harder than for events where it is detected higher up at 91–94 km.

The soft precipitation sentence has been removed.

Line 195. 'stays slightly elevated'. State the value of the electron density that is slightly elevated.

The elevated electron density value has been stated.

Line 213. Suggest change 'The light precipitation' to 'This modest increase in precipitating flux' or similar.

Changed accordingly

Lines 229–235. The results section discusses correlations between electron density and airglow temperature / intensity in Figure 3. Regression lines for the data need to be added to the scatter plots and the correlation values stated.

Regression lines and correlation coefficients have been added.

5. Discussion

Line 245. '10 degrees' should be '10 K'. Rather than 'often larger...', it should be stated for exactly how many events the temperature change was larger than 10 K.

Changed accordingly

Lines 247–248. 'the fast temperature decrease and equally quick recovery shown by the superposed epoch analysis (bottom left panel of Figure 3) would not be seen as a change in hourly or daily averaged epoch'. I agree that daily averaged epochs are unlikely to show the temperature decrease, but hourly averages may do so. Suggest the authors either remove the statement about hourly data or substantiate the claim e.g., by showing the 30-minute data smoothed to hourly averages.

The claim for these changes not being visible in the hourly data has been removed.

6. Conclusions

Line 302. Suggest change 'downward temperature decrease by 5–10 K/km at the airglow altitude' to 'decreasing temperature of 5–10 K/km over the airglow altitude range', or similar.

Changed as suggested

235 Line 310–311. 'Energetic electron precipitation' is defined as 'EPP' whereas that abbreviation was used previously (lines 1 and 26) for 'energetic particle precipitation'.

Changed to energetic particle precipitation here too.

Line 317. 'only a few decrease'. I think this should be 'only a few degrees'.

True, this has been changed to degrees.

240 Figure 1 The upper altitude of the electron density panel should be set at 160 km as there are no plotted data higher up.

The altitude range has been cut off at 160 km.

Figure 2 The colour bars of the electron density panels are missing titles.

Titles have been added to the colour bars.

245 The vertical axis limits of the electron density, temperature, and intensity panels need to be adjusted to more clearly show the plotted data without significant white space, and without clipping the error bars.

We prefer having the plots on the same scales for quicker comparison. This is one reason to show an example event with different axes ranges in Figure 1.

250 Figure 3 'The lower percentile for intensity (is) 35%.' Why is the lower percentile set at 35% rather than 25% as for temperature?

The band brightness during the first and second event was extremely low (100–200), and the 25% percentile is sensitive to that resulting in large point-to-point fluctuations which do not describe the bulk behaviour. To exclude the dominance of these two events we changed the percentile to 35% which is more descriptive for the rest of the events. This should have been mentioned in the first place but is mentioned in the new version.

255

'Each 30 min epoch time bin contains 6–10 temperature (intensity) values.' Why does the number of values in each bin vary between 6 and 10?

The variation in the number of data points is due to individual missing data points. As seen in Figure 2 we do not have 100% coverage of the temperatures and intensities. This is mentioned in the revised text.

260

---

## Author Response (AR4)

[revised manuscript text omitted]

**Response to Referee 5**

We thank Referee 5 for reading and evaluating the manuscript. All minor corrections have been implemented according the reviewer's suggestions.